# Get Set or Get Distracted? Disentangling Content-Priming and Attention-Catching Effects of Background Lure Stimuli on Identifying Targets in Two Simultaneously Presented Series

**DOI:** 10.3390/brainsci9120365

**Published:** 2019-12-11

**Authors:** Rolf Verleger, Kamila Śmigasiewicz, Lars Michael, Laura Heikaus, Michael Niedeggen

**Affiliations:** 1Department of Neurology, University of Lübeck, 23538 Lübeck, Germany; k.smigasiewicz@gmail.com (K.Ś.); a.l.heikaus@gmail.com (L.H.); 2Institute of Psychology II, University of Lübeck, 23538 Lübeck, Germany; 3Laboratoire de Neurosciences Cognitives, Aix-Marseille Université, CNRS, 13331 Marseille, France; 4Department of Psychology, Medical School Berlin, 12247 Berlin, Germany; lars.michael@medicalschool-berlin.de; 5Department of Pedagogy and Psychology, Free University of Berlin, 14195 Berlin, Germany; michael.niedeggen@fu-berlin.de

**Keywords:** rsvp, lure stimuli, priming, ERPs, N2pc

## Abstract

In order to study the changing relevance of stimulus features in time and space, we used a task with rapid serial presentation of two stimulus streams where two targets (“T1” and “T2”) had to be distinguished from background stimuli and where the difficult T2 distinction was impeded by background stimuli presented before T1 that resemble T2 (“lures”). Such lures might actually have dual characteristics: Their capturing attention might interfere with target identification, whereas their similarity to T2 might result in positive priming. To test this idea here, T2 was a blue digit among black letters, and lures resembled T2 either by alphanumeric category (black digits) or by salience (blue letters). Same-category lures were expected to prime T2 identification whereas salient lures would impede T2 identification. Results confirmed these predictions, yet the precise pattern of results did not fit our conceptual framework. To account for this pattern, we speculate that lures serve to confuse participants about the order of events, and the major factor distinguishing color lures and digit lures is their confusability with T2. Mechanisms of effects were additionally explored by measuring event-related EEG potentials. Consistent with the assumption that they attract more attention, color lures evoked larger N2pc than digit lures and affected the ensuing T1-evoked N2pc. T2-evoked N2pc was indistinguishably reduced by all kinds of preceding lures, though. Lure-evoked mesio-frontal negativity increased from first to third lures both with digit and color lures and, thereby, might have reflected expectancy for T1.

## 1. Introduction

Knowledge about factors limiting visual perception is important to understand how we become aware of the world around us. This issue has often been studied by means of the rapid serial visual presentation task (RSVP) where target stimuli are embedded in background stimuli presented rapidly one after the other. A well-investigated phenomenon is the “attentional blink”, which is the difficulty in discerning T2, the second of two target stimuli (“T1” and “T2”) when presented at some critical moment after T1 [1,2]. The present study investigates another handicap to identifying T2: Some background stimuli may resemble T2. Such “lure” stimuli, differing from the actual T2 only by occurring before T1 rather than afterwards, have been shown to impair T2 identification [3,4,5,6,7,8,9] and may be considered a model case for studying the time-dependent changes of relevance which the many objects in our environment may undergo, because the very same object that is irrelevant when encountered too early may become relevant later on. Several studies have provided evidence that the process launched by perception and associated rejection of lures is inhibition of detecting the targets [4,5,6,7,8,9,10,11].

In the studies by Niedeggen and colleagues [4,5,6,7,8,9] participants had to detect a color change of the fixation point to red (T1) and then immediately detect or discriminate the direction of coherent motion (T2) of dots surrounding fixation or, in [7], a flip of surrounding bars. Lags from T1 to T2 varied between 0 ms and 700 ms. Performance was nearly perfect for all measured lags (Figure 1 in [6]). Yet T2 identification drastically deteriorated at short T1–T2 lags (most at a lag of 0 ms) when lure stimuli were presented before T1. This negative effect was cumulative, increasing with the number of lures in a trial [5,9], well in line with the notion of a gradual increase of central inhibition [8].

In Zhang et al.’s studies [10,11], T1 was a red letter in a series of white letters at screen center, and T2 was a white digit, either also presented at center [10,11] or left or right [10]. Lures (i.e., white digits presented at center before T1) reduced T2 identification.

In Harris et al.’s study [12] a stream of object drawings was presented at screen center. Participants had to identify the two red objects (T1 and T2) among the other, black objects. The black object that was presented two frames before T1 could be the same object as the red T2. These lures had negative effects on T2 identification when T2 was presented briefly after T1. These negative effects turned to stable positive effects when lures and T2 were rotated or mirrored relative to each other.

In order to embed such time-dependent changes into a spatial dimension, in a previous study, we presented lure stimuli in dual-stream RSVP. In this task, first implemented by Holländer et al. [13] and used in our lab in several studies (from [14] onwards), two streams are presented, left and right from fixation, with T1 and T2 each occurring in either of these streams. In our standard version, background stimuli are black letters, T1 is a red letter, and T2 is a black digit. In our lure study in this set-up [15], lures, presented in half of the trials, were black digits like T2. This is similar to Zhang et al.’s [10,11] stimuli, except that they did not present two streams. In our paradigm with two streams, evidence was found for both positive and negative priming exerted by lures on T2 identification, formally similar to Harris et al.’s [12] results. Specifically, in our task [15] lures had negative effects on T2 identification when the lag between T1 and T2 was 3 frames, This effect was shifted towards positive priming when lures and T2 were in the same stream and when the intervening T1 was “out of the way” in the other stream and when one of three lures was identical to T2. This latter result apparently differs from the above-reported results by Niedeggen et al. and Harris et al. [4,5,6,7,8,9,12] where identity of lures and T2 impeded rather than facilitated T2 identification.

Because of these conflicting results, here we reasoned that both mechanisms might be effective, either one being triggered by specific relations between lures and T2. To detail, similarity of lures to T2 might result in positive priming of lure contents for discerning T2 [12,15] whereas the property of lures to capture attention due to their similarity to T2 might inhibit and impede T2 identification. In the present study, we aimed at separating these two presumed effects by assigning two different features to T2 and using two types of lures: Unlike in previous studies [10,11,15], the T2 targets stood out not only by being digits but also by their color, being blue rather than black. In different trials, lures were either three same-category lures, i.e., black digits, or three attention-catching, salient lures, i.e., blue letters. We assumed that digit lures will positively prime T2 identification, whereas salient lures, being only superficially similar to T2, will not be able to prime T2 identification but will rather distract attention and, thereby, impede T2 identification. To optimize the presumed distinction of effects, one of the three digit lures in a trial was identical to T2, like in half the lure trials of our previous study [15]. Positive priming from lures on T2 might be considered trivial when one of the lures is T2. But negative identity priming from lures on T2 has been shown as well [12], and moreover, negative priming from identical irrelevant to relevant stimuli when separated by masks or other stimuli has often been demonstrated [16,17,18].

Sample trials of the three conditions no-lures, digit lures, color lures are depicted in Figure 1.

### 1.1. Stream Change from Lures to T2

The supposed attention-distracting effect of salient color lures may involve a strong spatial component. I.e., the impeding effect of salient lures on T2 identification may become evident only if lures and T2 are in different streams, because lures will have attracted attention to their stream which then turns out to be the incorrect one. Correspondingly, no negative effect, or even some positive effect on T2 identification, is expected if lures and T2 are in the same stream because, then, the lures will have attracted attention to the correct stream.

In contrast, by being content-dependent, the supposed positive priming effect of digit lures on T2 identification might be less tightly linked to spatial conditions. Thus, this positive effect should not differ much between stream change and stream continuity from lures to T2.

### 1.2. Role of T1 Stream

The preceding considerations were made without taking the relevant and attention-catching stimulus between lures and T2 into account, which is T1, the red letter. In [15], some evidence was obtained for the assumption that positive priming effects built up by lure presentations are broken and even inverted by T1 presentation (signaling that the preceding digits were too early to be T2). To keep things simple, it may at least be expected here that the spatial position of T1 will have effects on spatially specific effects: The attention-distracting effects of salient lures on other-stream T2 might be enhanced by T1 occurring in the lure stream and attenuated by T1 occurring in the T2 stream.

### 1.3. Event-Related Potentials

Event-related potentials (ERPs) were recorded in order to provide additional evidence on differential processing of same-category digit lures and salient color lures. This additional evidence was expected to be particularly relevant to evaluate immediate processing of the lures because there were only indirect behavioral measures of lure processing (by their moderating effects on discerning T2 and T1).

### 1.4. Lure-Evoked ERPs

When a trial included lures, there were always three of them, occurring in the same stream. If lures evoke inhibition then this inhibition should accumulate across the three lures. Lure-evoked ERPs had been measured by Niedeggen et al. [8,9,17], Verleger et al. [13], and Zhang et al. [6]. In all studies, a negative ERP component was evoked by lures at anterior scalp sites around 300 ms. In Niedeggen et al.’s studies [7,8,19], this component became larger across the three lure positions within a trial. (Zhang et al. [10] presented only one lure per trial at a fixed position, immediately before T1). Not occurring in control conditions (where lures did not share features with the target), the increase was interpreted by Niedeggen et al. [7,8,19] as reflecting frontal “gating” against being prematurely activated by these stimuli. However, this increase was not replicated in Verleger et al.’s [15] study. To accommodate this divergence, we assume for the present study that such frontal negativity will not build up with the same-category digit lures but will do so with the distracting salient color lures.

In addition, our use of bilateral streams allowed for measuring the N2pc component. N2pc is a negative deflection around 250 ms after onset of laterally presented relevant stimuli, at contralateral sites above the visual cortex [20,21] reflecting spatially selective processing of these stimuli [22,23,24]. Measuring N2pc evoked by lures provides a test for the assumption that color lures will attract more attention than digit lures: If this is true, N2pc amplitudes are expected to be larger with color lures than with digit lures, particularly with the first of the three lures because when the first color lure has attracted attention to its side the focus of attention may stay on that side such that the further lures do not require a shift of attention any more.

### 1.5. T1-Evoked ERPs

Being a relevant and salient lateral event, the red-colored T1 evokes N2pc in this dual-stream task [12,23]. One might speculate that the preceding sequence of three lures is helpful to expecting T1 precisely in time, so T1-evoked N2pc might increase in lure trials. However, no such effect was obtained in our previous study [13]. What is to be expected, though, is that the shifting of attention by preceding color lures will affect T1-evoked N2pc. Without lures, attention will be distributed across the two streams. Color lures will attract attention to their stream. T1 will require less shifting of attention when presented in the same stream as color lures and will require more shifting of attention when color lures had occurred in the other stream. Accordingly, T1-evoked N2pc is expected to be smaller than in trials without lures when color lures and T1 are in the same stream, and to be larger when color lures and T1 are in different streams.

### 1.6. T2-Evoked ERPs

Likewise, being a relevant event, T2 will evoke N2pc, too [10,14,25,26]. In Zhang et al.’s study [10], N2pc was delayed when lures preceded. In contrast, in Verleger et al.’s study [15], N2pc amplitudes were reduced, but only when lures and T2 were in the same stream. It may be speculated that the possibly inhibition-related effect of [10] might become evident in the present study after color lures while the possibly priming-related effect in [15] might become evident after digit lures.

To summarize, the major purpose of the present study was to obtain divergent effects of the two types of lures on T2 identification, and the ERPs evoked by lures, T1, and T2 were expected to be helpful in describing the mechanisms of these lure effects.

## 2. Materials and Methods

### 2.1. Participants

Eighteen students (9 male) of the University of Lübeck participated. This sample size was well in the range of our previous studies using this task where consistent behavioral and ERP effects had been obtained. Informed written consent was obtained and 7 € per hour were paid. All participants reported normal or corrected-to-normal vision, normal color vision, and no history of neurological disorders. Four of those 18 participants turned out to be unable to identify T2 reliably above chance, with mean identification rates markedly below 20%, and were excluded from analysis. The remaining 14 participants (7 males) were aged 20 to 29 years (M = 25, SD = 2.7). One was left-handed (laterality score −100), the others were right-handed, with a mean score of +94 (SD 9) in the Edinburgh Handedness Inventory [27].

### 2.2. Stimuli and Apparatus

The task is illustrated in Figure 1. Two simultaneous sequences of black capital letters of the Latin alphabet were rapidly (9/s) presented left and right from fixation. The 17’’ screen had a white background (120 cd/m^2^) and was driven with 100 Hz at about 1.2 m from participants’ faces. Letters were 8.5 mm wide and 11 mm high (0.5° × 0.6° visual angle) with their inner edges 10 mm from fixation (0.6°). Fixation was marked by a small red cross (0.1° × 0.1°) at screen center. In each trial, two targets had to be identified. The first target (T1) was a red letter (24 cd/m^2^; D, F, G, J, K, or L). The second target (T2) was a blue digit (18 cd/m^2^; 1, 2, 3, 4, 5, 6, 7, 8, or 9). Background stimuli consisted of all other letters in black (standard), or in blue (color lures) or of the digits 1–9 in black (digit lures). Presentation^®^ software, version 14.5, was used for experimental control (Neurobehavioral Systems Inc., Berkeley, CA, USA).

### 2.3. Procedure

Participants were seated in a comfortable armchair in a dimly lit room in front of the computer screen. Their task in each trial was to identify the red letter (T1) and the ensuing blue digit (T2).

Each trial started with onset of the fixation cross. The two simultaneous letter streams started 800 ms later. Each stimulus pair was presented for 110 ms, immediately followed by the next frame. Background pairs either consisted of two standard stimuli or, in lure trials, of a standard stimulus and a lure (in three background pairs preceding T1). The three lures were presented in the same stream, which was either on the same side as the subsequent T2 or on the other side. There were five conditions: Same-side (as T2) color lures, other-side color lures, same-side digit lures, other-side digit lures, and no lures; 150 trials of each condition were presented in random order. Standard stimuli were randomly selected with replacement from the letter set (with a restriction against immediate repetition). T1 and T2 were randomly selected from the target sets. One of the three digit lures, randomly selected, was identical to the forthcoming T2 digit (resembling the feature of color lures whose color was always identical to T2) and the other two digit lures were randomly selected from the T2 set. T1 and T2 each were presented on the left or right side together with a standard stimulus on the other side.

In order to avoid fixed temporal expectancies [28] time-points of lures, T1, and T2 were varied across trials. T1 was, on average, at position 12.0 (±1.4 SD; range 10–14). Lures (both color lures and digit lures) occurred between positions 3 and T1 − 2, on average at positions 4.2 (±1.3 SD), 6.5 (±1.7), 8.8 (±1.7), i.e., in temporal intervals of 2.3 frames on average = 250 ms, ending on average 3.2 frames (350 ms) before T1. T2 followed T1 with lags of 110 ms (Lag 1) or 330 ms (Lag 3). Five standard-letter pairs followed T2. Therefore, trial length varied between 16 frames (when T1 came at the 10th position and T1–T2 lag was 1) and 22 frames (when T1 came at 14th position and T1–T2 lag was 3). Out of the four possible side × lag relations of T1 and T2 (same side / other side × lag 1 / lag 3) the same-side lag-1 relation was omitted altogether, in order not to overly lengthen the task, because T2 identification rates are relatively uninformative in this condition, being always close to 100%, due to “lag 1 sparing” [15,29,30]. Thus, of the 150 trials within each of the five lure-T2 relations (same-side color lures, other-side color lures, same-side digit lures, other-side digit lures, no lures) 50 trials each, in random order, applied one of the three T1-T2 relations, half of them with T2 on the left and half with T2 on the right: Same side with lag 3 (in shorthand notation *L3L* and *R3R*, denoting T1 side, lag, T2 side), side change with lag 1 (*R1L* and *L1R*), side change with lag 3 (*R3L* and *L3R*).

A standard keyboard was placed directly in front of participants on an adjustable base. At the end of each trial, 2.5 s after onset of the first stimulus-frame, participants were prompted by a message on the screen to enter their responses on the keyboard, first the T1 letter on the middle row and then the T2 digit on the number pad. When not knowing the answer, they had to guess. The next trial started immediately after the T2 response.

The five (lure conditions) × three (T1-T2 relations) × two (T2 sides) × 25 trials (=750 trials) were presented in random sequence, with a break after 375 trials. Before the task proper, some trials were presented in slow motion for practice, with 500 ms rather than 110 ms presentation rate.

### 2.4. Analysis of T1 and T2 Identification Rates

Separately for T1 and T2, percentages of trials with correct responses were computed in each of those five × three × two = 30 cells of the design. In order to avoid having an ill-defined three-level factor that would combine variation of lag (1 vs. 3) and of T1-T2 relation (same side vs. other side) we decided to focus analysis on the variation of same vs. other side in lag 3 data. To handle the five levels of the factor Lure Condition, analysis proceeded in two steps. First, no-lure data will be described, with respect to effects of target side (left, right) and of other-target side (same as target, other than target), with “target” and “other target” denoting T1 or T2 depending on analysis. Second, the effects of the lures were tested by subtracting the no-lure data from data of each of the four conditions with the lures, thereby extracting the net effects of lures in each of these conditions, and entering these differences to ANOVAs with the repeated-measurement factors Lure Type (digit, color) and Lure Side (same as target, other than target), additionally to Target Side and Other-Target Side. In additional analyses, the effects of Lag (1 vs. 3) were tested by comparing the other-target-on-other-side data between lag 3 and lag 1. These analyses had the factors Target Side and Lag for no-lure data, and Lure Type, Lure Side, Target Side, and Lag for lure effects. Only effects of the Lag factor will be reported from these analyses.

### 2.5. EEG Recording and Pre-Processing

EEG was recorded with Ag/AgCl electrodes (Easycap, www.easycap.de) from 60 scalp sites, which were 8 midline positions from AFz to Oz and 26 pairs of symmetric left and right sites. Further electrodes were placed at the nose-tip for off-line reference and at Fpz as connection to ground. On-line reference was Fz. For artifact control, the electrooculogram (EOG) was recorded, vertically (vEOG) from above vs. below the right eye and horizontally (hEOG) from positions next to the outer tails of the eyes. Voltages were amplified from DC to 250 Hz by a BrainAmp MR plus, A–D converted, and stored with 500 samples/s per channel. Off-line processing was done with Brain-Vision Analyzer software (version 2.03). Data were re-referenced to the nose-tip and low-pass filtered at 20 Hz (Butterworth zero phase filters, attenuation of 12 dB / octave). Then 600 ms epochs, starting 100 ms before the respective event, were cut out of the EEG for analyzing the potentials evoked by each of the three lures, by T1, and by T2. These epochs were edited for artifacts, rejecting trials with voltage differences at any recording site that exceeded 150 µV or exceeded 40 µV between successive data points. Mean values of 100 ms pre-stimulus epochs were subtracted, and data were averaged over trials separately by conditions.

For lures, separate averages were formed for left and right 1st, 2nd, and 3rd digit and color lures. Control averages were formed from equivalent positions in no-lure trials and subtracted from the lure epochs. (Lure positions had been assigned by the sequence-generating program also in no-lure trials).

Analysis of T1 and T2 EEG epochs was restricted to trials where T2 followed with a lag of three frames because potentials evoked by T1 and T2 were inextricably mixed when T1 was followed by the other-stream T2 with lag 1. For T1, trials with correct response to T1 were selected, and separate averages were formed over trials for left and right T1 in the five lure conditions (no lures, digit lures in the same stream as T1 and in the other stream than T1, color lures in the same stream as T1 and in the other stream than T1). For T2, trials with correct responses to both T1 and T2 were selected, and separate averages were formed over trials for left and right T2 in the five lure conditions (no lures, digit lures in the same stream as T2 and in the other stream than T2, color lures in the same stream as T2 and in the other stream than T2). Unlike for T1 analysis, this was done separately for trials with T1 and T2 on same sides and T1 and T2 on different sides.

To obtain contralateral–ipsilateral (con–ips) differences of each symmetric left-right pair of recording sites, the left-site average was subtracted from the right-site average when the event (lure, T1, or T2) was left (e.g., PO8 − PO7), vice versa when the event was right (e.g., PO7 − PO8) and the mean of these two con–ips differences was formed. Grand means over participants were calculated for illustrating the results. Con–ips differences were also formed for hEOG and these hEOG difference waveforms were inspected for systematic deviations from baseline within 500 ms after lure onsets, indicating eye movements toward the lures. It was found that only two of the 14 participants had notable deviations after the first color lure. These deviations amounted to about 5 µV only, corresponding to an average eye movement of about 0.3° towards the lure stream, which we considered small enough to keep these participants in the sample. Including those two participants, the grand average waveforms of hEOG deviations toward lures reached a maximum of about 1 µV at 400 ms after onsets of 1st and 2nd lures, and much smaller values after 3rd lures, with 1 µV corresponding to average eye movements of about 0.07° towards the lure stream, which still appeared to be an acceptable level.

As will be detailed in the Results section, parameters were determined in the |PO7 − PO8| con–ips difference waveforms evoked by lures (minus no-lures), by T1, and by T2, as well as in current source densities evoked by lures (minus no-lures) recorded at FCz. Statistical analysis was performed by ANOVA with repeated measurements. *P* values of effects of the three-level factors Lure Position and Epoch (as defined in the Results section) will be reported after Greenhouse–Geisser correction.

## 3. Results

### 3.1. Target Identification

Percentages of trials in which the targets were identified are displayed in Figure 2. Additionally, mean values and standard deviations are compiled in the Appendix A. Each data point was computed from 25 trials per participant.

#### 3.1.1. T1: Trials without Lures

T1 was identified in 84% of no-lure trials on average (black line in the upper left panel of Figure 2). No-lure identification rates were submitted to ANOVA with the factors T1 Side (left, right) and T2 Side (same as T1, other than T1). T1 was identified better when the following T2 was in the same stream and T1 was on the right (R*3R* in Figure 2, with absence of italics denoting the target under consideration, here T1), as indicated by a main effect of T2 Side, *F*_1,13_ = 9.4, *p* = 0.009, and the interaction of T2 Side × T1 Side *F*_1,13_ = 4.5, *p* = 0.05, resolved to an effect of T2 Side for right T1, *F*_1,13_ = 11.8, *p* = 0.004, and no such effect for left T1, *F*_1,13_ = 1.2, n.s.

Additionally, the other-side-than-T2 data were compared between lag 3 and lag 1 in an ANOVA with the factors T1 Side (left, right) and Lag (1, 3). No effect was significant, all *F*_1,13_ ≤ 2.1, *p* ≥ 0.17.

#### 3.1.2. Lure Effects on T1 Identification

Lure effects were tested by subtracting the no-lure data from data of each of the four conditions with lures and entering these differences to ANOVAs. The main ANOVA was conducted on lag 3 trials with the repeated-measurement factors Lure Type (digit, color) and Lure Side (same as T1, other than T1), additionally to T1 Side and T2 Side. ANOVA results are compiled in Table 1.

Color lures had larger negative effects on T1 identification than digit lures (blue vs. grey in Figure 2), Lure Type *F*_1,13_ = 5.1, *p* = 0.04, and lure effects were more negative when the following T2 was on the same side as T1 (L*3L* and R*3R*) than when sides changed (L*3R* and R*3L*), T2 Side *F*_1,13_ = 9.1, *p* = 0.01.

When lures, T1, and T2 all were in one stream (solid blue and grey lines in Figure 2), right-side T1 was more affected by lures than left-side T1 (*r*R*3R* > *l*L*3L*): Lure Side × T1 Side × T2 Side *F*_1,13_ = 7.5, *p* = 0.02, resolved to an effect of T1 Side × T2 Side of *F*_1,13_ = 6.7, *p* = 0.02, for lures on the same side as T1 (in contrast to *F*_1,13_ = 1.0, n.s., for lures on other side than T1) and further, for these same-side lures, to an effect of T1 Side of *F*_1,13_ = 10.1, *p* = 0.007, when T2 was on same side as T1 (in contrast to *F*_1,13_ = 0.7, n.s., for T2 on the other side than T1).

Finally, the significant four-fold interaction (T1 Side × Lure Side × Lure Type × T2 Side *F*_1,13_ = 6.9, *p* = 0.02) reflected two separate effects, namely, first, the outlying positive value of the dashed grey line in Figure 2 at R*3L* and, second, the outlying negative value of the dashed blue line at L*3L*. To detail: The outlying positive value of the dashed grey line at R*3L* was reflected by resolving the fourfold interaction to threefold interactions of Lure Type × T1 Side × T2 Side separately for same-side and other-side lures, which yielded a significant result for other-side lures (dashed lines in Figure 2), *F*_1,13_ = 14.2, *p* = 0.002, in contrast to *F*_1,13_ = 0.9, n.s., for same-side lures. Then resolving that threefold interaction for other-side lures to effects of T1 Side for each of the four combinations of Lure Type × T2 Side yielded a significant effect of T1 Side for digit lures and other-side T2 (dashed grey line: R*3L* more positive than L*3R*), *F*_1,13_ = 9.1, *p* = 0.01, in contrast to *F*_1,13_ ≤ 0.7, n.s., for each of the other three combinations. Second, the outlying negative value of the dashed blue line at L*3L* was reflected by resolving the overall fourfold interaction to threefold interactions of Lure Side × T1 Side × T2 Side separately for color lures and digit lures which yielded a significant result for color lures, *F*_1,13_ = 12.5, *p* = 0.004, in contrast to digit lures, *F*_1,13_ = 0.0, n.s. Resolving that threefold interaction for color lures to effects of Lure Side for each of the four combinations of T1 Side × T2 Side yielded a significant effect of Lure Side for L*3L*—other side (dashed blued line in Figure 2) more negative than same-side (solid blue line)—, *F*_1,13_ = 8.9, *p* = 0.01, in contrast to the other three combinations of T1 Side × T2 Side (R*3R*, L*3R*, R*3L*), *F*_1,13_ ≤ 2.5, *p ≥* 0.15, n.s.

A second ANOVA was run to compare other-side-than-T2 data between lag 3 (L*3R*, R*3L*) and lag 1 (L*1R*, R*1L*), replacing the previous factor T2 Side by Lag (lag 1, lag 3) to have the factors Lure Type, Lure Side, T1 Side, and Lag. No effect of Lag reached significance, all *F*_1,13_ ≤ 2.8, *p* ≥ 0.12.

#### 3.1.3. T2: Trials without Lures

T2 was identified in 61% of no-lure trials on average. Similar to T1 analysis, ANOVA on no-lure rates of T2 identification in lag 3 trials had the factors T2 Side (left, right) and T1 Side (same as T2, other than T2). Left T2 was identified better than right T2, *F*_1,13_ = 7.5, *p* = 0.02. T2 on the same side as T1 was better identified than T2 on the other side, *F*_1,13_ = 10.3, *p* = 0.007. These two factors did not interact, *F*_1,13_ = 0.3, n.s.).

Additionally, the lag 1 data were compared to the other-side-than-T1 data of lag 3 in an ANOVA with the factors T2 Side (left, right) and Lag (1, 3). T2 tended to be better identified after Lag 3 than after Lag 1, *F*_1,13_ = 4.3, *p* = 0.06. The advantage of left T2 over right T2 was again significant, *F*_1,13_ = 5.4, *p* = 0.04, and did not interact with Lag, *F*_1,13_ = 0.3, n.s.

#### 3.1.4. Lure Effects on T2 Identification

Like with T1, the main ANOVA on differences between lure and no-lure trials was conducted on lag 3 trials, with the factors Lure Type (digit, color) and Lure Side (same as T2, other than T2), additionally to T1 Side (same as T2, other than T2) and T2 Side (left, right). ANOVA results are compiled in the top half of Table 2.

As predicted, color lures were much more detrimental than digit lures, *F*_1,13_ = 27.5, *p* < 0.001. Besides, the pattern of effects differed much between color lures and digit lures (Figure 2), as reflected by five out of seven possible interactions of Lure Type being significant (top half of Table 2). Therefore, separate ANOVAs were computed for the two lure types, with the factors Lure Side, T1 Side, and T2 Side (bottom half of Table 2).

Color lures had an overall negative effect (constant term of ANOVA differing from zero: *F*_1,13_ = 22.0, *p* < 0.001). As predicted, this effect was larger when lures were in the other stream than when they were in the same stream (Lure Side: *F*_1,13_ = 5.9, *p* = 0.03). Furthermore, the negative effect was largest on *L3*L, i.e., left T2 preceded by left T1 (T1 Side × T2 Side *F*_1,13_ = 5.4, *p* = 0.04; simple effect of T1 Side on left T2 *F*_1,13_ = 5.3, *p* = 0.04; on right T2 *F*_1,13_ = 0.1, n.s.).

Digit lures had an overall zero effect (constant term of ANOVA *F*_1,13_ = 0.2, n.s.). But effects differed between lure sides, *F*_1,13_ = 9.3, *p* = 0.009, and between T1 sides, *F*_1,13_ = 15.9, *p* = 0.002, in both cases being negative when sides (of lures or of T1) were the same as T2 and positive when sides differed from T2. Important were the strong interactions of Lure Side × T2 Side, *F*_1,13_ = 25.4, *p* < 0.001, and of Lure Side × T1 Side, *F*_1,13_ = 8.1, *p* = 0.01. Both interactions reflected that there were large moderating effects (of T2 Side and of T1 Side) on lure effects on T2 identification when lures and T2 were in the same stream (solid grey line in Figure 2; effect of T2 Side *F*_1,13_ = 16.2, *p* = 0.001; of T1 Side *F*_1,13_ = 29.7, *p* < 0.001) in contrast to absence of effects when lures and T2 were in different streams (dashed grey line; effect of T2 Side *F*_1,13_ = 2.0, *p* = 0.19; of T1 Side *F*_1,13_ = 2.0, *p* = 0.18). Thus, when lures and T2 were in the same stream (solid grey line), lure effects were negative both for left T2 (*L3*L and *R3*L) and when T1 was on the same side (*L3*L and R*3R*), and were more positive both for right T2 and when T1 was on the other side. Thereby, when looking at these interactions from the viewpoint of differential effects of lure side, these effects of lure side—negative with same-side lures, positive with other-side lures—were focused on left T2 (effect of Lure Side on left T2 *F*_1,13_ = 43.6, *p* < 0.001; on right T2 *F*_1,13_ = 1.1, n.s.) and on same-side T1 (effect of Lure Side with same-side T1 *F*_1,13_ = 26.5, *p* < 0.001; with other-side T1 *F*_1,13_ = 0.0, n.s.). Whether these effects were indeed significantly different from zero was tested by evaluating the deviations from zero of the constant terms of ANOVAs conducted on single levels of those two two-way interactions Lure Side × T2 Side and Lure Side × T1 Side. Effects were reliably negative when T2 was in the same stream as lures (solid grey line) and either was left (*L3*L and *R3*L), *F*_1,13_ = 16.8, *p* < 0.001, or when T1 was in the T2 stream (*L3*L and *R3*R), *F*_1,13_ = 23.9, *p* < 0.001. Effects were reliably positive when T2 was in the other stream than lures (dashed grey line) and was left (*L3*L and *R3*L), *F*_1,13_ = 5.7, *p* = 0.03, and when lures were in the same stream as T2 (solid grey line) but T1 was in the other stream (*R3*L and *L3*R), *F*_1,13_ = 7.5, *p* = 0.02.

Additionally, the lag 1 data were compared to the other-side-than-T1 data of lag 3, replacing the previous factor T1 Side by Lag (lag 1, lag 3) to have the factors Lure Type, Lure Side, Lag, and T2 Side. No effect of Lag came below the *p* = 0.05 threshold (all *F*_1,13_ ≤ 4.5, *p* ≥ 0.054).

### 3.2. ERP Reflections of Lure Processing

Lure-evoked ERPs were computed from 150 trials per participant (minus artifact-affected epochs).

#### 3.2.1. Lure-evoked N2pc

Lure-evoked contralateral–ipsilateral differences from posterior–lateral sites |PO7–PO8| are displayed in Figure 3. There is obviously a negative component (N2pc) peaking around 250 ms with the 1st lure. N2pc was measured by computing mean amplitudes at 200–300 ms and submitting these values to ANOVA with the factors Lure Type (color, digit) and Serial Position (1st, 2nd, 3rd lure).

Overall, N2pc was largest with the 1st lure (Serial Position: *F*_2,26_ = 12.4, *p* = 0.001) and larger with color than digit lures, *F*_1,13_ = 4.8, *p* = 0.048. Both factors tended to interact, *F*_2,26_ = 3.4, *p* = 0.07. Indeed, in separate analyses for each serial position, Lure Type was significant at the first position only. *F*_1,13_ = 5.5, *p* = 0.04 (*F*_1,13_ ≤ 2.0, n.s., for 2nd and 3rd positions). In separate analyses of the two lure types, the Position effect was significant in either analysis, though being somewhat larger with color lures (*F*_2,26_ = 9.2, *p* = 0.004) than with digit lures (*F*_2,26_ = 6.1, *p* = 0.009).

It may be suggested that the early negative deflections in the 2nd position waveforms, peaking at about 170 ms, are accelerated N2pc peaks. But a second peak is visible at about 250 ms in these waveforms as well. Given that attention was already shifted towards the lure stream because of the first lure, we agree with one reviewer of this paper that the negativity may reflect increased N1 components evoked by this attended lure stream (e.g., [31]) rather than an early N2pc. When quantifying this early activity as mean amplitudes 150–200 ms, there was no effect of Lure Type, *F* ≤ 2.1, n.s., but a distinct effect of Position, *F*_1,13_ = 11.7, *p* < 0.001, with largest values for the 2nd position.

#### 3.2.2. Frontal Activity

The upper panel of Figure 4 shows lure-evoked ERPs at FCz. A large negative peak seems to be evoked by the first color lure at about 300 ms. However, the scalp map of these data suggests that this peak may stem from volume-conducted posterior negativity. In contrast, negative peaks with fronto-central focus are visible with 2nd and 3rd lures, though at later latencies. In order to enhance the weights of location-specific activity and get rid of volume conducted posterior negativity, current source densities (CSDs) [32] were computed (lower panel of Figure 4). Indeed, as the maps in Figure 4 show, current-sinks had a focus at or near FCz, with their amplitudes appearing to vary between serial positions: negligible activity with the first lure, distinct but relatively late activity with the second lure (400–500 ms after lure onset), somewhat earlier activity (300–400 ms) with the third lure.

To quantify these impressions, an ANOVA was computed on mean CSD amplitudes of the 300–400 ms and 400–500 ms epochs, with the factors Epoch (early, late), Serial Position (1st, 2nd, 3rd lure), Lure Type (digit, color), and Lure Stream (left, right). The main effect of Serial Position (*F*_2,26_ = 3.8, *p* = 0.047) and the interaction of Epoch × Serial Position (*F*_2,26_ = 3.9, *p* = 0.043) were further explored by computing pairwise ANOVAs on lure #1 vs. #2 and on #2 vs. #3 separately for the two epochs. Negativity increased from lure #1 to #2 in the late epoch (*F*_1,13_ = 6.6, *p* = 0.02; early epoch: *F*_1,13_ = 0.6, n.s.) and from lure #2 to #3 in the early epoch (*F*_1,13_ = 7.3, *p* = 0.02; late epoch: *F*_1,13_ = 0.1, n.s.). Different from what we had expected, there was hardly any difference between digit and color lures. The only effect of Lure Type was a moderation of the just-mentioned Epoch × Serial Position interaction, Epoch × Serial Position × Lure Type *F*_2,26_ = 4.6, *p* = 0.02, which appeared to reflect a tendency with 3rd lures where negativity tended to decrease from early to late epoch with digit lures, *F*_1,13_ = 4.3, *p* = 0.06, not so with color lures, *F*_1,13_ = 0.0, n.s. There were no reliable differences between left and right streams (all effects of Lure Stream *F*_1,13_ ≤ 3.6, *p* ≥ 0.08).

### 3.3. ERP Reflections of Lure Effects on T1 Processing

T1-evoked ERPs were computed from about 120 trials per participant (150 trials per cell minus about 20% trials with incorrect responses) minus artifact-affected trials.

T1-evoked contralateral–ipsilateral differences from posterior–lateral sites |PO7–PO8| are displayed in Figure 5. There is a large N2pc, peaking at about 220 ms. Lure effects were measured by computing the differences between lure trials and no-lure trials in the mean amplitudes at 175–275 ms for each of the four lure conditions and submitting these difference values to an ANOVA with the factors Lure Type (color, digit) and Lure-T1 Relation (same side, other side).

T1-evoked N2pc reliably increased when color lures had preceded on the other side. (Main effect of Lure-T1 Relation *F*_1,13_ = 19.8, *p* = 0.001, modified by the interaction of Lure-T1 Relation × Lure Type *F*_1,13_ = 5.9, *p* = 0.03, resolved to a simple effect of Lure-T1 Relation with color lures, *F*_1,13_ = 55.8, *p* < 0.001, and absence of such effect with digit lures, *F*_1,13_ = 0.3, n.s.). The main effect of Lure Type was not significant (*F*_1,13_ = 1.0, n.s.).

Confirming the ANOVA results, *t* tests of the four lure conditions against no lures yielded a significant increase of negativity for color lures in the other stream, *t* = −4.6, *p* < 0.001, and no significant differences for any of the three other conditions, *t* ≤ |1.2|, *p* ≥ 0.24.

### 3.4. ERP Reflections of Lure Effects on T2 Processing

T2-evoked ERPs were computed from 28 trials per participant (50 trials per cell minus about 33% trials with incorrect responses minus artifact-affected trials).

T2-evoked contralateral–ipsilateral differences from posterior–lateral sites |PO7–PO8| are displayed in Figure 6, separately for T2 on the same side as T1 and T2 on the other side from T1. Since T2 followed on T1 with a lag of 3, = 330 ms, the leftmost time-point of Figure 6 (−100 ms) is time-point 230 ms of Figure 5 (except that Figure 5 also includes trials where T2 followed T1 at lag 1 whereas Figure 6 includes trials with lag 3 only). At that time, T1-evoked N2pc just reaches its peak in Figure 5. Thus, Figure 6 starts with the decreasing slope of T1-evoked N2pc. This decrease is plotted with same polarity (descending towards positivity) when T2 and T1 are on the same side (left panel of Figure 6) and with opposite polarity in the other case (ascending towards negativity; right panel).

There is a distinct N2pc when T2 was preceded by T1 in the other stream (right panel of Figure 6), peaking at about 250 ms, and a similar though apparently smaller component is visible when T2 was preceded by T1 in the same stream (left panel of Figure 6). To quantify these impressions, N2pc was measured by computing mean amplitudes 200–300 ms.

First, T2-evoked N2pcs were compared between same-side T1 and other-side T1 in an ANOVA on no-lure trials with the one factor T1-T2 Relation (same side, other side). Indeed, N2pc was larger after other-side T1 than after same-side T1, *F*_1,13_ = 13.8, *p* = 0.003. Then, lure effects were measured by computing the differences between lure trials and no-lure trials, separately for each of the four lure conditions and for T1-T2 same-side and other-side trials, and submitting these difference values to an ANOVA with the factors Lure Type (color, digit), Lure-T2 Relation (same side, other side) and T1-T2 Relation (same side, other side). In this ANOVA on difference values, the constant term was different from zero, *F*_1,13_ = 7.2, *p* = 0.02, indicating that there was an overall N2pc difference between lure and no-lure trials which, as Figure 6 shows, was an overall reduction of N2pc by the presence of lures. None of the factors had significant influence on this lure effect, though, all *F*_1,13_ ≤ 2.8, *p* ≥ 0.12. When nevertheless testing each of the eight lure conditions separately against the no-lure condition, reductions were significant when digit lures preceded on the same side and also T1 was on that side (*t* = 2.3, *p* = 0.04; solid grey line in left panel of Figure 6) and when color lures preceded on the same side and T1 was on the other side (*t* = 2.7, *p* = 0.02; solid blue line in right panel of Figure 6).

## 4. Discussion

In a demanding task with spatially varying rapid presentation, we tested the impact of early occurrence of features that were typical of the second target. The original variation of the present study from several earlier studies [3,4,5,6,7,8,9,10,11,12,15] was that lures resembled T2 in two different, alternative ways. Our major expectation about these two types of lures was confirmed: Their effects on T2 identification grossly diverged (main effect of Lure Type *p* < 0.001). T2 identification was impeded by color lures and was mainly improved by digit lures.

We had assumed as a mechanism underlying this divergence of effects that color lures falsely attract attention to their stream while digit lures positively prime the identification of digits. From this viewpoint, the obtained results will be discussed in detail. It will be concluded that these mechanisms were indeed at work but cannot account for the whole pattern of results.

### 4.1. T2 Identification

We focused analysis on the lag 3 trials. This was an a posteriori decision (suggested by one reviewer of this paper) because the three-level factor T1-T2 Relation in the ANOVA on all data was hard to interpret, being composed of the distinction between same-side and different-side T1 at lag 3 and the distinction between lags 3 and 1 for different-side T1. By dropping the lag 1 data, analysis could focus on the effects of same-side vs. different-side targets.

This mixed three-level factor had been a consequence of our decision to omit lag 1 same-side T1-T2 from the experiment altogether because T2 identification uses to be near perfect in this condition and otherwise the experiment would have been too long. In the light of this a posteriori decision to drop the remaining lag 1 condition from the main analysis, in planning the experiment it might have been wiser to achieve the necessary reduction of trials not by complete omission of lag 1 same-side trials but rather by balanced reduction of lag 1 trials both for same-side and different-side targets.

In any case, corresponding to the analysis in the Results section, also the following discussion will focus on the lag 3 trials.

#### 4.1.1. No-Lure Trials

The mean identification rate of 66% in no-lure trials at lag 3 was lower than we had expected based on preceding studies where T2 identification was frequently close to 80% for lags > 1 [33]. A problem in designing this study had been that T2 stood out from its background not only by being a digit among letters, as usual, but additionally by its color, being blue among black stimuli. From pilot data, we worried that identification rates would be too much improved by this additional feature, reaching ceiling, therefore we increased the presentation rate from our usual 7.7/s (130 ms per frame) to 9/s (110 ms per frame). This might have been too fast.

Alternatively, identification rates might have suffered even in no-lure trials from the presence of lures. This appears plausible because T2 identification rates were low not only in the present study but also in our previous lure study [15] where mean rates in no-lure trials at lag 3 barely reached 60% although T2 appearance and presentation rates conformed to our usual standard that has usually resulted in rates of 80%. Perhaps the presence of potentially distracting lure stimuli causes participants to invest more attention in the task than usual. Such increased investment of attention might actually be detrimental to performance [34,35]. Or the concepts may be applied that have been suggested by episodic distinction approaches to the attentional blink [2,18]: Having expected the irrelevant lures first which were not to be included in the “evaluation window” [18], participants might have still waited for those events in the no-lure trials and might have opened their “evaluation window” too late for processing the actually relevant events.

Left-side T2 was better identified than right-side T2, as has been found again and again in dual-stream RSVP tasks [11,12,33]. Until recently, we had ascribed this to dominance of the right hemisphere in shifting attention. But recent studies made a clear case for learned strategies of left-to-right reading as the underlying cause [36,37,38].

#### 4.1.2. Lure Effects

As predicted, color lures were much more detrimental than digit lures. Different from our predictions, though, were the spatial specificities of those effects.

For color lures, we had predicted that their negative effect would be most evident when lures and T2 are in different streams and that there might be even a positive effect on same-stream T2. Indeed, color lures’ negative effects on T2 identification were smaller when T2 was in the same stream than when T2 was in the other stream. Yet same-stream lures still had negative effects rather than positive ones. Thus, while it might still be true that color lures attracted attention (cf. below, 4.3), as expected, this shift of attention did not appear to be the only mechanism responsible for their negative effects on T2 identification. Nor were these effects moderated by T1 position in the predicted way: We had assumed that the attention-distracting effects of color lures on other-stream T2 (dashed blue line in Figure 2) might be enhanced by T1 occurring in the lure stream and attenuated by T1 occurring in the T2 stream. This hypothesis was more incorrect than correct. In line with the hypothesis was the relatively large negative effect of other-stream (left-side) lures on *L3*R. But against expectation, the maximum negative effect of lures on other-stream T2 occurred when T1 and T2 were in the same stream (*L3*L).

For digit lures, we had predicted that spatial effects would be largely absent because the relevant mechanism of their presumed positive effects on T2 identification was assumed to be priming of digits as a category. Based on the actual results we cannot firmly exclude category priming as a mechanism. But the spatial specificity of the obtained effects came as a surprise. The expected positive effects on T2 identification were considerably more distinct when lures and T2 were in different streams than when they were in the same stream, and were restricted to effects of right-side lures (improving identification of left-side T2 in the sequences rR1L, rL3L, rR3L, and of right-side T2 in the sequence rL3R). The effect in the first three cases is remarkable because left-side T2s have already an advantage over right-side T2s in the no-lure control condition, so the positive lure effect did not serve for restoring the balance but rather exacerbated the differences.

Further below, we will offer a hypothesis about the mechanisms underlying these effects of color lures and digit lures. Since our original hypothesis failed, this proposed mechanism will remain speculative.

### 4.2. T1 Identification

#### 4.2.1. No-Lure Trials

T1 was identified best when presented in the right stream, followed by T2 in the same, right stream. The right-side advantage for T1 has been found in several of our studies using this task where T1 data for right and left sides were separately reported. Either as a main effect or interacting with T2 side or lag, the effect was obtained in Exp.1 of [14], in Taiwanese and Israelis in [38], and in [39,40,41], with a total of 163 participants. The effect was not obtained in some other studies of ours using this task ([25,42,43,44,45]) with a total of 123 participants. Similarly, an advantage for identification of T1 followed by T2 on the same side has been found several times, either as a main effect or interacting with T1 side or lag [12,15,40,46], with a total of 73 participants. Yet one study on 50 participants yielded the opposite effect [47] and several studies, with a total of 206 participants, yielded no effect [25,38,39,41,42,43]. Thus, the present good identification of right T1 followed by right T2 seems to be a both replicable and variable phenomenon.

The advantage of right T1 might be related to left-hemisphere specialization for language [48], and its variability across experiments may confirm Hellige’s [49] conclusion that laterality of letter identification depends on the detailed circumstances. Accounting for the effect of side of the following T2 on identification of the previous T1 seems more challenging. Worse T1 identification when T2 is on the same side [47] might be due to backward masking. The opposite better T1 identification when T2 is on the same side [14,15,40,46] may be taken to suggest that, like a post-cue [50,51], T2 is able to draw attention not only to its ongoing stimulus stream but, given beneficial circumstances, also to the short-term memory representation of previous stimuli in this stream.

#### 4.2.2. Lure Effects

We had not stated any hypothesis about effects of lures on T1 identification.

Based on our initial attention-attraction hypothesis of color lures, it makes sense to assume that T1 identification was impaired when color lures had preceded in the other stream, withdrawing attention from the stream in which T1 was presented. As Figure 2 showed (blue dashed line), this did indeed apply to those two of the four *lag 3* instances where T1 and T2 were in the same stream (L*3L* and R*3R*) but did not apply when T2 was in the other stream (L*3R* and R*3L*). Moreover, this hypothesis does not account for the large negative effect on T1 identification both for color and digit lures when, lures, T1 and L2 were all in the same, right-side stream. Nor is there any clue why the was positive priming on T1 when digit lures were left, T1 was right, and T2 was left again (grey dashed line in Figure 2 at R*3L*).

### 4.3. Lure-Evoked ERPs: N2pc

The N2pc component is a marker of shifts of spatial attention [20,21,22,23,24], possibly in order to individuate the attention-attracting stimuli [52]. Based on the notion of attentional shifts, it was assumed that color lures will evoke larger N2pc than digit lures and, more trivially, that the first lure in a row will evoke larger N2pc than the second and third ones because the three lures are all on the same side (as in [15]). Both assumptions were confirmed. These results confirm that color lures attracted more attention than digit lures.

In recent studies, a contralateral positive potential, following the N2pc or even instead of N2pc, has been described to be the major ERP signature of distracting stimuli in visual search arrays [23]. It has been suggested that this “P_D_” indicates an inhibitory mechanism to prevent stimuli from catching attention [53]. The question may, therefore, be asked why lures evoked contralateral negativity (N2pc) rather than positivity (P_D_). As pointed out by one reviewer of this manuscript, one reason might be that lures were, by definition, similar to targets. Such similarity has resulted in N2pc rather than P_D_ being evoked by distractors in previous studies (e.g., [54,55]). A P_D_ is typically observed with targets that are easy to find and clearly distinct from the distractor [23,56,57] whereas an N2pc will result, like in the present study (as well as in [10,15]), when targets are difficult to find [58].

### 4.4. Lure-Evoked ERPs: Frontal Negativity

In previous studies, lures evoked a negative component at 300 ms [7,8,10,15,19]. This negativity became larger across the three lure events within a trial in [15] and was accordingly interpreted by those authors as reflecting frontal gating for inhibition. That increase was not replicated in [15] (and could not be replicated in [10], because one lure only was presented in each trial). To accommodate these divergent results, we assumed that frontal gating is only necessary when lures are potentially harmful to T2 identification. Therefore, it was predicted that such frontal negativity will increase over lure repetitions mainly with color lures.

Genuinely fronto-central negative activation (excluding the peak evoked by first color lures at 300 ms which was volume-conducted from posterior sites) did occur in the present data, as corroborated by computing CSDs. But this activation occurred later than in most of those previous studies (except [8]), from 340 ms onwards, and took the form of slow shifts.

As expected, frontal negativity increased across lure repetitions. But unexpectedly, this increase occurred with digit lures as well as with color lures whereas we had expected that the effect would be larger with color lures because only color lures had negative effects on T2 identification. Interpreting frontal negativity as indicator of gating for inhibitory purposes [8,10] we would have to conclude that the positively priming digit lures underwent inhibition to the same degree as the distracting color lures. This does not seem probable.

As an alternative, it may be suspected that these late potentials evoked by the later lures actually are not evoked by lures but rather by the ensuing T1. (The minimum interval between 3rd lure and T1 was 220 ms only, and 330 ms from 2nd lure to T1. Thus, although mean intervals were much longer, 350 ms and 600 ms, there might have remained some vestige of T1 in the analyzed epochs). However, any such effects had been prevented by having subtracted data of no-lure control trials from the lure trials. Thereby, any T1-evoked potentials were subtracted out. This leaves us to assume that these components were evoked by the later lures specifically in expectancy of T1, i.e., that these FCz-focused relatively slow potentials are CNVs (Contingent Negative Variations, [59,60,61]) in expectancy of T1 because the third and second lures provided some temporal structure to the otherwise undifferentiated stream of background stimuli, thereby enabling participants to build some short-term expectations. To conclude, these slow frontal negative shifts might rather be CNV-type expectancy potentials for better localizing T1 in time than reflect inhibitory activity.

### 4.5. T1-Evoked N2pc

T1-evoked N2pc was expected to increase to the extent that an attention shift is required (e.g., [46]). Without lures, attention will be randomly distributed across the two streams whereas color lures will attract attention to their stream. T1, presented on average 350 ms after the final lure, will, therefore, require either less or more shifting of attention depending on its being presented either in the same stream as color lures or in the other stream. Accordingly, T1-evoked N2pc was expected to be either smaller or larger than in trials without lures depending on color lures and T1 being either in the same stream or in different streams.

Results were in reasonably good agreement with these predictions: T1-evoked N2pc was not at all affected by digit lures, was increased when color lures preceded in the other stream and showed a weak tendency to be reduced (far from significance, though) when color lures preceded in the same stream.

Additionally, we had speculated that T1-evoked N2pc might increase after any lures because the sequence of three lures is helpful to expecting T1 precisely in time. Evidence for this supportive function of lures was provided by the CNV-type negativity evoked by the final lures as discussed in the previous section. However, amplitude of T1-evoked N2pc was not generally increased by preceding lures. (Nor was it in our previous lure study, [15]). In fact, we do not know of evidence showing that N2pc is larger when evoked by events whose timing can be better predicted. Thus, this hypothesis might have been unfounded.

### 4.6. T2-Evoked N2pc

We had expected that, by using the two types of lures, our data might reconcile conflicting results of previous studies. Specifically, color lures might lead to a delay in N2pc, reflecting ongoing inhibition, like in Zhang et al. [10], and digit lures might lead to a decrease of N2pc amplitude when lures and T2 are in the same stream, like in Verleger et al. [15].

However, this is not what we obtained. Rather, the ANOVA yielded a general reducing effect of preceding lures on N2pc amplitude. Close inspection of Figure 6 suggests that this may be particularly true, like in [15], when lures were presented on the same side as T2, and indeed this is where significant deviations from no-lure trials were found in *t* tests on single conditions. But this was as well true for digit lures (as predicted) as for color lures (not predicted). In any case, differences of lure effects between lure types and lure sides were not significant in the ANOVA, casting doubt on any differential interpretation other than that lures generally reduced T2-evoked N2pc.

Such general reduction of T2-evoked N2pc amplitudes after preceding lures needs some interpretation. Above, with N2pc evoked by lures and by T1, we have interpreted N2pc reductions as consequence of the fact that attention had already been shifted before to the respective location, which is compatible with a facilitating effect of lures. On the other hand, N2pc may be reduced because participants are less capable of shifting their attention (e.g., [26]) which is compatible with an interfering effect of lures. For these reasons, with the different lure effects on N2pc not reliably differing from each other, the interpretation of this general reduction remains unclear.

This lack of specificity in lure effects might well be a Type-2 error caused by noisy signals. By design, each of the 10 analyzed conditions (no lures, digit and color lures on same and on other side × T2 on same vs. other side as T1 at lag 3) consisted of 50 trials, namely 25 left-T2 and 25 right-T2 trials. This is certainly at the lower limit of what is required for N2pc. Moreover, this number of 50 trials was reduced by the fact that only correctly responded T2 trials entered analysis which were, e.g., for the no-lure condition, about 66% of the lag 3 trials (cf. Table 1). Together with rejection of trials for artifacts in the EEG, this led to a mean number of 28 trials per condition.

### 4.7. Speculation on Underlying Mechanisms

It appears that the ERP results may serve for defining some boundary conditions: Indeed, color lures attracted more attention than digit lures (lure-evoked N2pc), both types of lures might have been equally helpful in localizing T1 in time (lure-following frontal negativity), color lures indeed made a shift of attention necessary when T1 was in the other stream (T1-evoked N2pc), and lures somehow affected attention shifts to T2 although the nature of that affection remained unclear (T2-evoked N2pc). However, it seems that the unexpected result pattern of lure effects on target identification cannot be brought into a coherent picture by means of these ERP results. Thus, we are left to speculate, based on the pattern of the above-discussed identification effects and on previous literature.

Building on perceptual-episode accounts of the attentional blink [2,18] it appears to us that the major function of lures might be to make participants uncertain about the order of events. Unlike in the tasks with multiple targets explored by Wyble et al. [2] and Dell’Acqua et al. [62], T1 can hardly be mistaken to be T2 in our task (one being a letter, the other a digit). But, if sufficiently similar to T2, lures may fool participants into thinking that one of the lures might have already been T2 and that, therefore, T1 had been missed. When then T1 is nonetheless encountered, it might get processed (after the lure which would act as first target) as a second target in time, thereby suffering from the asymmetry of T2 identification, being much worse identified when presented on the right than when presented on the left. This agrees with the cumulation of negative effects of lures on T1 identification in the R*3R* sequence. In order to account for the fact that this is particularly true with R*3R*, and not equally with R*3L* and R*1L*, we have to additionally assume that the presentation of T2 in the same stream as T1 (R*3R*) is apt to increase participants’ confusion about the roles assigned to the successive targets.

The crucial factor in the difference between color and digit lures might have been which one is more similar to T2, thereby confusing participants more about the order of events. We may assume that, in our task, participants searched for T2 by its color above all, perhaps in order to use the same strategy as in searching for T1 which was exclusively defined by its color (red letter among black letters). This might be the major reason why color lures had a negative impact on T2 identification: Blue letters, more than black digits, might have led participants to believe that the blue digit had already been presented, thereby terminating their search for the true T2. This reasoning may account for the minor role of spatial specificity of color-lure effects. In fact, similar conclusions have emerged from a research tradition where lures were not similar to T2 but rather to the only target that had to be detected in a central RSVP stream, and lure-type distractors were placed in the periphery of this central stream: Also here, color lures in the periphery had spatially non-specific negative effects, by interfering with detection of the centrally presented target [63,64,65,66,67,68].

In contrast to color lures, the black digit lures could hardly be mistaken for T2 and, thereby, could serve as positive primes for T2 identification. Perhaps a certain extent of confounding the lures and T2 was still possible when digit lures and T2 were in the same stream. This might explain why positive priming was regularly obtained when digit lures and T2 were in different stream and not when they were in the same stream. Finally, the reason for the asymmetry of the positive digit-lure effect (exclusively by right-side lures and mainly on left-side T2) is not really clear. It may be due to better processing of the alphanumeric character of lures in the left hemisphere or, alternatively due to the preferred processing of left-side T2 in this dual-RSVP paradigm.

## 5. Conclusions

In conclusion, the spatial variation enabled by the dual-stream task and the use of two types of lures provided rich opportunity for observing a variety of lure effects which require more solid theoretical interpretation than could here be achieved post hoc.

## Figures and Tables

**Figure 1 brainsci-09-00365-f001:**
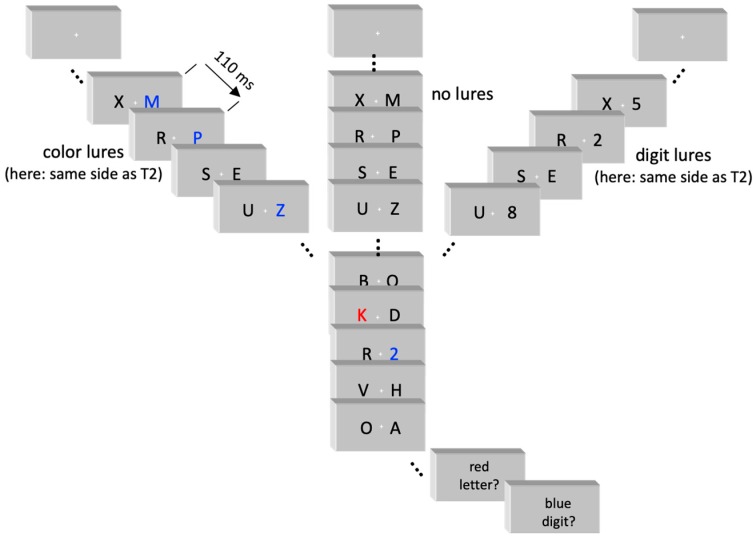
Sequence of events in a trial. Participants had to identify the red letter (T1) and the blue digit (T2) embedded in a stream of background stimuli. At least nine pairs of background stimuli were presented before T1. In 80% of trials, three of these pairs contained stimuli that resembled T2 (“lures”), either by their blue color or by their being a digit. The three lures could be on the same side as the ensuing T2 (as in these examples) or on the other side.

**Figure 2 brainsci-09-00365-f002:**
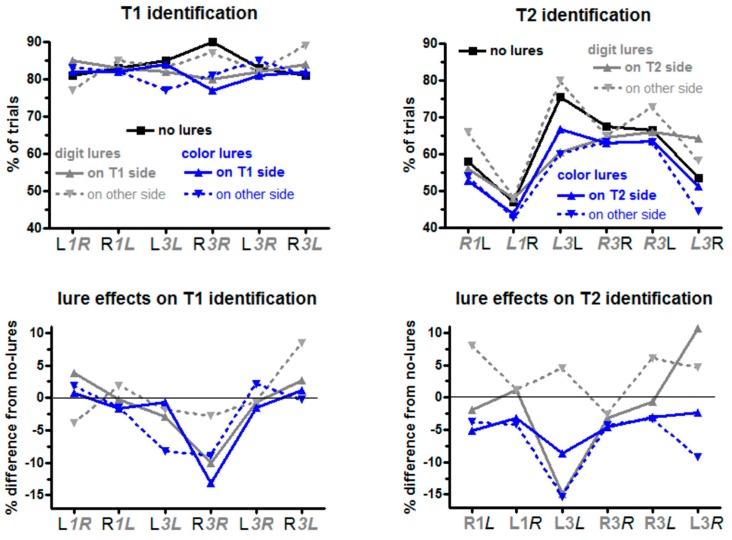
Lure effects on identification rates of T1 and T2. The upper panels display the percentages of trials in which T1 (left) and T2 (right) were identified. The lower panels display the differences between trials with lures from trials without lures. No-lure trials (upper panels only) are denoted by black lines, trials with digit lures by grey lines, and trials with color lures by blue lines. Trials where lures were in the same stream as the target (T1 and T2, respectively) are denoted by solid lines (grey and blue) and trials where lures were in the opposite stream are denoted by dashed lines. Both for T1 and T2, the six values on the x axes denote left-right-left-right-left-right targets. E.g., “L1R” is a trial where T1 was left and T2 was right (separated from each other by 1 frame). Thereby, L1R is the leftmost value for T1 (L*1R*) and the second value for T2 (*L1*R). The main ANOVAs were conducted on the lag 3 data (four rightmost values in each panel) and additional ANOVAs compared lag 1 and lag 3 other-target-on-other-side data (two leftmost and two rightmost values in each panel).

**Figure 3 brainsci-09-00365-f003:**
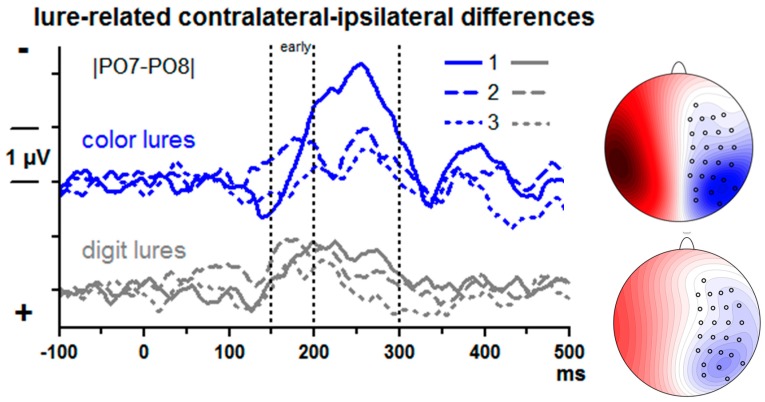
Contralateral–ipsilateral event-related potential (ERP) differences evoked by the lures. Data are grand means across participants, recorded from left and right posterior sites PO7 and PO8. Depicted are differences between lure-trials and corresponding epochs of no-lure trials. Unit on x-axis is milliseconds, time-point zero is lure onset. Unit on y-axis is microvolts, negative voltage is plotted upwards. Waveforms evoked by the 1st lure are shown as solid lines, by the 2nd lure as dashed lines, and by the 3rd lure as dotted lines. ERPs evoked by color lures and digit lures are plotted with blue and grey lines, respectively. The scalp maps show the view on the head (120°) from above. Recording sites (small circles) are depicted on one hemisphere only because the contralateral–ipsilateral differences were pooled across left and right sides. Blue is contralateral negativity, red is positivity. Scale is ±4 µV. Displayed are topographic distributions of N2pc evoked by the first lure at its peak latency: 256 ms with the first color lure, 224 ms with the first digit lure.

**Figure 4 brainsci-09-00365-f004:**
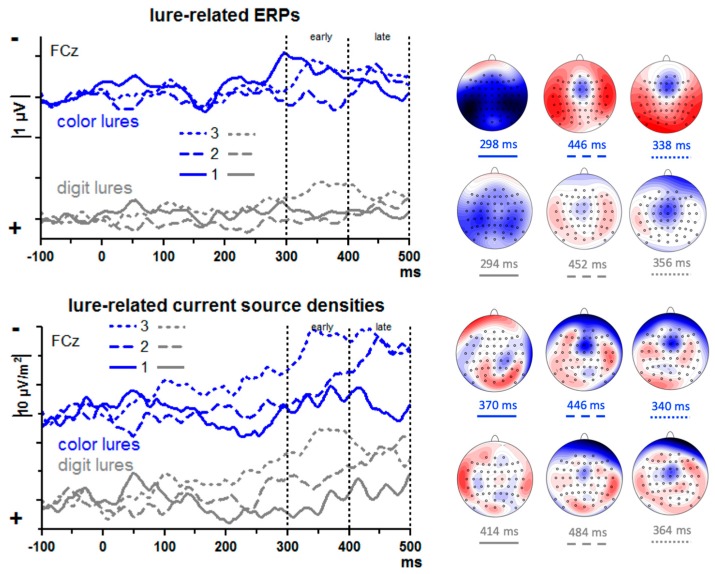
ERPs and current source densities evoked by the lures. Data are grand means across participants, recorded from the fronto-central midline site FCz. The upper panel displays the ERPs, and the lower panel displays current source densities, i.e., ERP data from which ERP data from surrounding sites were subtracted. Depicted are differences between lure-trials and corresponding epochs of no-lure trials. Unit on x-axis is milliseconds, time-point zero is lure onset. Unit on y-axis is microvolts in the upper panel and microvolts per square meter in the lower panel, negative values are plotted upwards. Waveforms evoked by the 1st lure are shown as solid lines, by the 2nd lure as dashed lines, and by the 3rd lure as dotted lines. Data evoked by color lures and digit lures are plotted with blue and grey lines, respectively. The scalp maps show the view on the head (120°) from above. Recording sites are denoted by the small circles. Displayed are topographic distributions at the indicated latencies for each of the six conditions. In the upper panel, blue denotes negative polarity, red positive polarity, and scale is ±2 µV. In the lower panel, blue denotes negative sinks, red denotes positive sources, and scale is ±15 µV/m^2^.

**Figure 5 brainsci-09-00365-f005:**
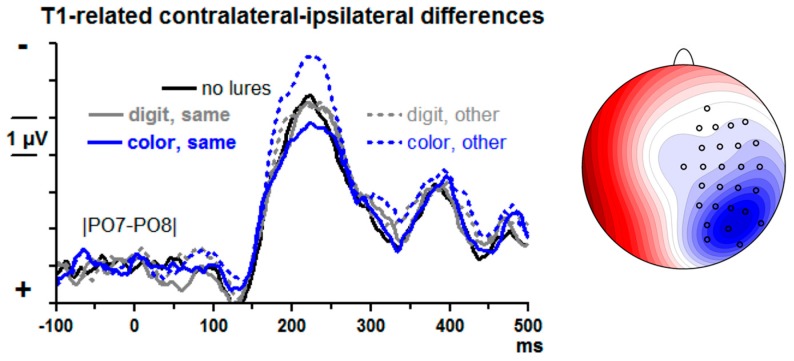
Contralateral–ipsilateral ERP differences evoked by T1. Data are grand means across participants, recorded from left and right posterior sites PO7 and PO8. Unit on x-axis is milliseconds, time-point zero is lure onset. Unit on y-axis is microvolts, negative voltage is plotted upwards. Waveforms evoked in no-lure trials are black, from color-lure trials blue, and from digit-lure trials grey. Trials where lures were in the same stream as T1 are denoted by solid lines (grey and blue) and trials where lures were in the opposite stream are denoted by dashed lines. The scalp map shows the view on the head (120°) from above. Recording sites (small circles) are depicted on one hemisphere only because the contralateral–ipsilateral differences were pooled across left and right sides. Blue is contralateral negativity, red is positivity. Scale is ±7 µV. Displayed is the topographic distribution of N2pc evoked in no-lure trials at the peak latency of N2pc (222 ms).

**Figure 6 brainsci-09-00365-f006:**
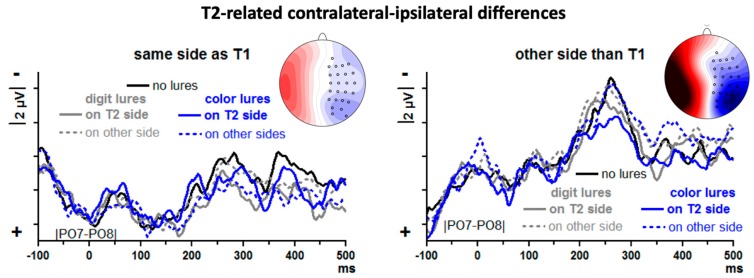
Contralateral–ipsilateral ERP differences evoked by T2. Data are grand means across participants, recorded from left and right posterior sites PO7 and PO8. Unit on x-axis is milliseconds, time-point zero is lure onset. Unit on y-axis is microvolts, negative voltage is plotted upwards. Waveforms evoked in no-lure trials are black, from color-lure trials blue, and from digit-lure trials grey. Trials where lures were in the same stream as T2 are denoted by solid lines (grey and blue) and trials where lures were in the opposite stream are denoted by dashed lines. The left panel displays data where T1 was in the same stream as T2, and the right panel displays data where T1 was in the other stream. Time-point −100 ms is approximately the peak of that preceding T1. The scalp maps show the view on the head (120°) from above. Recording sites (small circles) are depicted on one hemisphere only because the contralateral–ipsilateral differences were pooled across left and right sides. Blue is contralateral negativity, red is positivity. Scale is ±8 µV. Displayed is the topographic distribution of N2pc evoked in no-lure trials at the peak latencies of N2pc (282 ms in same-stream-as-T1 trials, 260 ms in opposite-stream trials).

**Table 1 brainsci-09-00365-t001:** ANOVA on effects of lures (deviations of lure conditions from the no-lure condition) on T1 identification rates.

Effect	*F* (*p*)	Effect	*F* (*p*)
T1 Side (left, right)	0.4	T2 Side × Lure Side	0.2
T2 Side (same as T1, other than T1)	**9.1 (0.01)**	Lure Type × Lure Side	4.0 (0.07)
Lure Type (digit, color)	**5.1 (0.04)**	T1 Side × T2 Side × Lure Type	0.9
Lure Side (same as T1, other than T1)	2.3	T1 Side × T2 Side × Lure Side	**7.5 (0.02)**
T1 Side × T2 Side	3.9 (0.07)	T1 Side × Lure Type × Lure Side	0.6
T1 Side × Lure Type	3.1 (0.10)	T2 Side × Lure Type × Lure Side	2.9
T1 Side × Lure Side	3.8 (0.07)	T1 Side × T2 Side × Lure Type × Lure Side	**6.9 (0.02)**
T2 Side × Lure Type	0.2		

Degrees of freedom are 1,13 throughout. *F* and *p* values are printed in bold when *p* ≤ 0.050. *p*-values were entered when *p* ≤ 0.10.

**Table 2 brainsci-09-00365-t002:** ANOVA effects on lure effects (deviations of lure conditions from the no-lure condition) on T2 identification rates: main analysis followed by separate ANOVAs for the two lure types.

Effect	*F* (*p*)	Effect	*F* (*p*)
T1 Side (same as T2, other than T2)	**5.7 (0.03)**	T2 Side × Lure Side	**8.3 (0.01)**
T2 Side (left, right)	1.4	Lure Type × Lure Side	**10.4 (0.007)**
Lure Type (digit, color)	**27.5 (<0.001)**	T1 Side × T2 Side × Lure Type	**5.5 (0.04)**
Lure Side (same as T1, other than T1)	1.2	T1 Side × T2 Side × Lure Side	0.6
T1 Side × T2 Side	0.9	T1 Side × Lure Type × Lure Side	3.5 (0.08)
T1 Side × Lure Type	**13.9 (0.003)**	T2 Side × Lure Type × Lure Side	**8.6 (0.01)**
T1 Side × Lure Side	**4.7 (0.050)**	T1 Side × T2 Side × Lure Type × Lure Side	**5.0 (0.04)**
T2 Side × Lure Type	0.2		
**Color Lures**		**Digit Lures**	
T1 Side (same as T2, other than T2)	1.3	T1 Side	**15.9 (0.002)**
T2 Side (left, right)	0.6	T2 Side	2.0
Lure Side (same as T1, other than T1)	**5.9 (0.03)**	Lure Side	**9.3 (0.009)**
T1 Side × T2 Side	**5.4 (0.04)**	T1 Side × T2 Side	0.2
T1 Side × Lure Side	0.0	T1 Side × Lure Side	**8.1 (0.01)**
T2 Side × Lure Side	0.0	T2 Side × Lure Side	**25.4 (<0.001)**
T1 Side × T2 Side × Lure Side	3.9 (0.07)	T1 Side × T2 Side × Lure Side	0.9

Degrees of freedom are 1,13 throughout. *F* and *p* values are printed in bold when *p* ≤ 0.050. *p*-values were entered when *p* ≤ 0.10.

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
