# Peer review of "Get Set or Get Distracted? Disentangling Content-Priming and Attention-Catching Effects of Background Lure Stimuli on Identifying Targets in Two Simultaneously Presented Series"

_brainsci, 2019, doi:10.3390/brainsci9120365_

Round 1

Reviewer 1 Report

The authors did a very good job answering my reviews, and I want to particularly commend their improvement of the results section which is much clearer. I agree that an additional behavioural experiment is too much to ask given the timetable set by the journal and the specific circumstances. 

The one minor point I want to raise again is the interpretation of the N2pc. Undoubtedly, the presence of an N2pc means that a spatial shift took place. However, since an N2pc can emerge even when the target’s location is known in advance (Craston, Wyble, Chennu & Bowman, JoCN 2019; Eimer & Kiss, JoCN 2008; Foster, Bsales & Awh, preprint), thus allowing participants to shift attention in advance obviating the need to shift, it seems inaccurate to claim that a reduction in the N2pc amplitude is necessarily related to worse shifting. At the same time, a reduction of the N2pc is expected even the N2pc is related to downstream spatially-specific attention selection (be it individuation, engagement, localization, or another spatially-selective process) and this does not seem to change the researchers conclusions. I think that raising the possibility (even very briefly) that the reduction in N2pc reflected a disruption in downstream attentional process would better reflect the literature.

Author Response

The authors did a very good job answering my reviews, and I want to particularly commend their improvement of the results section which is much clearer. I agree that an additional behavioural experiment is too much to ask given the timetable set by the journal and the specific circumstances. 

Response: Thank you for this evaluation!

The one minor point I want to raise again is the interpretation of the N2pc. Undoubtedly, the presence of an N2pc means that a spatial shift took place. However, since an N2pc can emerge even when the target’s location is known in advance (Craston, Wyble, Chennu & Bowman, JoCN 2019; Eimer & Kiss, JoCN 2008; Foster, Bsales & Awh, preprint), thus allowing participants to shift attention in advance obviating the need to shift, it seems inaccurate to claim that a reduction in the N2pc amplitude is necessarily related to worse shifting. At the same time, a reduction of the N2pc is expected even the N2pc is related to downstream spatially-specific attention selection (be it individuation, engagement, localization, or another spatially-selective process) and this does not seem to change the researchers conclusions. I think that raising the possibility (even very briefly) that the reduction in N2pc reflected a disruption in downstream attentional process would better reflect the literature.

Response: Motivated by this comment, the phrase in the Discussion (l.700) “The N2pc component is a marker of shifts of spatial attention [20-24]. Therefore, it was assumed …” was extended to read “The N2pc component is a marker of shifts of spatial attention [20-24], possibly in order to individuate the attention-attracting stimuli [52]. Based on the notion of attentional shifts, it was assumed …” (The new reference [52] is Mazza, Pagano, & Caramazza, JoCN 2013).

Besides, we do not agree with the reviewer on this issue. The literature quoted by the reviewer does not speak to this issue: Craston et al. (most probably their 2009 study is meant) did not report N2pc. Eimer & Kiss (2008) report N2pc evoked by cues whose locations were always unknown. In contrast, the study of Åšmigasiewicz, Hasan, & Verleger (JoCN 2017) reported T1-evoked N2pc and T2-evoked N2pc when T1 and/or T2 location is or is not disclosed at trial onset. When T1 location was announced before, T1-evoked N2pc decreased to almost 50% of its original size (their Figure 2) corresponding to the notion of attentional shifting. Similarly, when T2 location was announced before, T2-evoked N2pc decreased significantly (although only slightly).

Reviewer 2 Report

The authors satisfactorily addressed my concerns

Author Response

Thank you for your evaluation.

An acknowledgment to all three reviewers has been added to the manuscript.

Reviewer 3 Report

The paper have responded to my earlier concerns and I think the paper has improved. There are two minor issues the authors may want to consider.

Line 108: "of otherwise continuously randomly moving dots surrounding fixation" could be removed without losing much information. Readability would be better.

In my pdf, the tables had a header and a caption. Possibly, something went wrong in the formatting, but there should only be a header.

While the format of tables 1 and 2 is original and space-saving, it is not easy to understand. I would favor a simple vertical list of main effects and interactions with their F-values and p-values in the columns. Because is a multi-row table, it is also easy to confound F- and p-values. A more classical format would avoid this difficulty. Also, the label "main effects of column factors" is not entirely correct (as the authors acknowledge) because there is an interaction. Thus, to allow for appreciation of the ANOVA results at a single glance, I would favor a more traditional Table.

Author Response

The paper have responded to my earlier concerns and I think the paper has improved.

Response: Thank you for this evaluation. An acknowledgment to all three reviewers has been added to the manuscript.

There are two minor issues the authors may want to consider.

Line 108: "of otherwise continuously randomly moving dots surrounding fixation" could be removed without losing much information. Readability would be better.

Response: Done. Thank you for the suggestion.

In my pdf, the tables had a header and a caption. Possibly, something went wrong in the formatting, but there should only be a header.

Response: According to Brain Sciences guidelines, “All Figures, Schemes and Tables should have a short explanatory title and caption.” But the reviewer is entirely right insofar as the caption should not repeat the header. Which we changed accordingly.

While the format of tables 1 and 2 is original and space-saving, it is not easy to understand. I would favor a simple vertical list of main effects and interactions with their F-values and p-values in the columns. Because is a multi-row table, it is also easy to confound F- and p-values. A more classical format would avoid this difficulty. Also, the label "main effects of column factors" is not entirely correct (as the authors acknowledge) because there is an interaction. Thus, to allow for appreciation of the ANOVA results at a single glance, I would favor a more traditional Table.

Response: We followed the advice and provided a simple list for both tables which, in fact, did not need much more space than the previous version.

This manuscript is a resubmission of an earlier submission. The following is a list of the peer review reports and author responses from that submission.

Round 1

Reviewer 1 Report

This study examines the effect of exposure to distractor stimuli with features that resemble the target on performance in an attentional blink task. The study reexamines two lines of research showing that such stimuli can be either beneficial or detrimental to performance. The second target (T2) was defined as a blue digit, and participants had to report its identity. Prior to the first target (T1), participants were either exposed to black digits or to blue letters. The authors hypothesized that the blue letters would capture attention and therefore disrupt T2 identification whereas the black digits would prime the target and enhance T2 identification. This pattern was mostly observed, with some exceptions. The authors also expected a specific relationship  between the spatial positions of the distractors (lures) and T2 which was not confirmed.

This study is a sensible development to the authors’ previous studies. The authors should be applauded for clearly stating their hypotheses and the way in which they were not supported by the results. Moreover, I appreciate that the authors’ did not try to mask the complex pattern of results. Given the request for a time-limited review, I will not go into details about many minor concerns, but instead will detail broad issues which I think should be improved upon in a future revision prior to publication.

Presentation of the research questions: While the hypotheses were clearly stated, it was not clear from the introduction how these hypotheses are embedded within the broader literature about the effects of lures. For example, how was the conclusion that inhibition guides the disruptive effects of lures reached? What was the differences in the designs between this study and previous study? Without this information is it difficult to assess the contribution of the current study to the broader literature. Moreover, it was not clear to me whether the fact that the paradigm is an AB paradigm is important to the main research question (i.e., do lures only increase\reduce the blink?). I think the authors should give much more detail about the studies that were cited, and how the authors reached the hypothesis that some lures can prime T2 and others can disrupt T2.

Attentional capture by lures: the hypotheses and the conclusions drawn from the results rely to a large extent on the question of which lures produce attentional capture. The authors suggest that a colour-matching lure should be more salient and produce more capture than a category-matching lure. It is pretty safe to assume that participants used the colour as the selection feature and not the identity. As such, it is reasonable to assume that the colour would produce goal-driven attentional capture. I think this can be stated clearly (and put in the context of previous studies about attentional capture and the effects of lures).
The hypothesis that attentional capture by lures that are irrelevant but share the target’s color would reduce accuracy is supported by previous studies (Folk et al., 2002, 2008; Moore & Weissman, 2010, 2011; Zivony & Lamy, 2014, 2016). However, while the authors assume that attentional capture would have a spatially specific effect, these studies found that disruptive effect is not spatial (or at least not entirely spatial), as it didn’t matter if the capturing non-target was in the same stream as the target (Folk et al., 2008, Zivony & Lamy, 2016) or in a different stream (Folk et al., 2002; Moore & Weissman, 2010; Zivony & Lamy, 2014). Instead, it was interpreted that the presence of a cue can trigger an AB and disrupt the triggering of attentional engagement to subsequent targets. The larger disruption by colour lure (indicated, for example the reduced accuracy on T1) coupled with an absence of location-specific effects can surely be explained in this context (though the location specific effect of digit lures could not). Moreover, given that the lures themselves were temporally distant from T2, it is possible that their presence reduced the T2-locked N2pc and accuracy by means of increasing the time it took to process T1 (thereby increasing the depth of the blink created by T1).

References:

Folk, C. L., Leber, A. B., & Egeth, H. E. (2002). Made you blink! Contingent attentional capture produces a spatial blink. Perception & psychophysics64(5), 741-753.
Folk, C. L., Leber, A. B., & Egeth, H. E. (2008). Top-down control settings and the attentional blink: Evidence for nonspatial contingent capture. Visual Cognition16(5), 616-642.
Moore, K. S., & Weissman, D. H. (2010). Involuntary transfer of a top-down attentional set into the focus of attention: Evidence from a contingent attentional capture paradigm. Attention, Perception, & Psychophysics72(6), 1495-1509.
Moore, K. S., & Weissman, D. H. (2011). Set-specific capture can be reduced by pre-emptively occupying a limited-capacity focus of attention. Visual Cognition19(4), 417-444.
Zivony, A., & Lamy, D. (2014). Attentional engagement is not sufficient to prevent spatial capture. Attention, Perception, & Psychophysics76(1), 19-31.
Zivony, A., & Lamy, D. (2016). Attentional capture and engagement during the attentional blink: A “camera” metaphor of attention. Journal of experimental psychology: human perception and performance42(11), 1886.

Results section readability: This is a complex design with numerous analyses, and together with the abbreviated condition names, makes the results section somewhat difficult to follow. Some of the phrasing throughout the results section can be improved to make the reading more accessible. I would prefer to see the details about each ANOVA right before the results themselves, as I found myself going back and forth from methods to results. Another possible suggestion, which may or may not be realistic, is to present a figure that is collapsed across conditions that are less relevant to the interpretation of the main findings (given that the full results are already reported in the table).

Results validity: As the authors mention themselves in the discussion, the study may have problems of power, given the numerous conditions and small amount of trials per condition. This can lower the readers’ confidence in the validity and replicability of the results. For example, while the multiple comparisons between each data point and 0 is indeed the easiest way to interpret the effects of lures, it also raises the concern of alpha inflation, especially for results whose p-value are close to .04. Moreover, given that many of the interaction effects were significant in ways that were not predicted by the authors (many of them only barely significant), it is very hard to conclude which of them reflect a real effect and which is spurious. This can be remedied in another behavioural experiment which aims to replicate the main results.

N2pc results: when testing the N2pc for lures, the N2pc for the 2nd and 3rd lures may be reduced because of an AB effect (as reported by numerous previous studies, including from the main authors’ group as well as in other studies). This makes the analysis of the 2nd and 3rd lure less informative. Moreover, given that attention was already shifted towards the first lure, the negativity may reflect SSVEPs towards the attended stream rather than a real N2pc (e.g., Müller et al., 1998). This would explain why the 2nd lure has a different latency. To reject this explanation, the authors can show that the non-lure distractors do not produce an enhanced negativity. If the authors wish to show latency differences, I would prefer to see an analysis of latency onset rather than analysis of an early time window that was selected by looking at the results. In general, I think the N2pc analysis should be in the same time window throughout the manuscript, as different time windows raises the readers’ concern that they were tailored to the results. Regarding the lures, can the authors specify which of lures N2pc were different from 0?

Reference:
Müller, M. M., Picton, T. W., Valdes-Sosa, P., Riera, J., Teder-Sälejärvi, W. A., & Hillyard, S. A. (1998). Effects of spatial selective attention on the steady-state visual evoked potential in the 20–28 Hz range. Cognitive Brain Research, 6(4), 249-261.

Discussion:

Inhibition account: given that digit lures and colour lures produced the same enhanced negativity in FCz, and the finding that the effect of digit lures is generally positive, what evidence is there for the inhibition account? At the very least at seemed that enhanced negativity from one lure to another does not reflect enhanced negativity.

As the authors state, the complex pattern of results does not allow for a clear conclusion about the effects of lures. Here, as well, an additional behavioral experiment may be beneficial. For example, to test the possibility that the lures are used to predict the targets’ temporal position, two different blocks of temporal predictability vs. no predictability can be introduced.

Minor issues:

Introduction:
1. the authors could go into more details about why the study is important.
2. I’m not sure that the lures used in this study reflect time-dependent changes of relevance.
3. The N2pc does not reflect attentional shifting per se, but rather a downstream attentional process (Kiss, Van Velzen & Eimer, 2008; Zivony, Allon, Luria & Lamy, 2018), and can occur even when no attentional shifting is required. It would be helpful to present the evidence that the N2pc is smaller when attention was already captured to the target’s side. This is also an issue that affects the conclusions that can be drawn from the N2pc findings.

Method:
1. If I understand correctly, the targets were much brighter than the contralateral distractors. This should be addressed as a limitation, as the effects may be related to perceptual imbalances rather than attentional imbalances.
2. Personally I found the boldface font somewhat distracting. Perhaps Italics would be better?

Results:
1. In table 1: the headline should be T2, rather than target. It will be helpful to have a decimal added to the averages reported and SEs (preferably within-subject SEs) instead of SDs, which can ease the interpretation of the table.
2. In figure 2, lower panels: perhaps add a dotted line on 0 to help reader orient themselves.
3. Sometimes lag 1 and lag 3 are written as lag1 and lag3.
4. Given that many of the effects were significant, the authors may consider using a table to detail the results of the main ANOVA for lure effects on T1 and T2, and then use the main text to give further details and refer to the table.
5. “In this ANOVA, the constant term was different from zero … indicating that N2pc was overall reduced by the presence of lures” – I don’t understand how the ANOVA’s constant leads to that conclusion…

Author Response

Reviewer 1

This study examines the effect of exposure to distractor stimuli with features that resemble the target on performance in an attentional blink task. The study reexamines two lines of research showing that such stimuli can be either beneficial or detrimental to performance. The second target (T2) was defined as a blue digit, and participants had to report its identity. Prior to the first target (T1), participants were either exposed to black digits or to blue letters. The authors hypothesized that the blue letters would capture attention and therefore disrupt T2 identification whereas the black digits would prime the target and enhance T2 identification. This pattern was mostly observed, with some exceptions. The authors also expected a specific relationship between the spatial positions of the distractors (lures) and T2 which was not confirmed.

This study is a sensible development to the authors’ previous studies. The authors should be applauded for clearly stating their hypotheses and the way in which they were not supported by the results. Moreover, I appreciate that the authors’ did not try to mask the complex pattern of results. Given the request for a time-limited review, I will not go into details about many minor concerns, but instead will detail broad issues which I think should be improved upon in a future revision prior to publication.

Our general response: We thank this reviewer very much for his/her insightful comments. In fact, I have rarely encountered such constructive and insightful comments from all contributing reviewers as in the present case.

Presentation of the research questions: While the hypotheses were clearly stated, it was not clear from the introduction how these hypotheses are embedded within the broader literature about the effects of lures. For example, how was the conclusion that inhibition guides the disruptive effects of lures reached? What was the differences in the designs between this study and previous study? Without this information is it difficult to assess the contribution of the current study to the broader literature. Moreover, it was not clear to me whether the fact that the paradigm is an AB paradigm is important to the main research question (i.e., do lures only increase\reduce the blink?). I think the authors should give much more detail about the studies that were cited, and how the authors reached the hypothesis that some lures can prime T2 and others can disrupt T2.

Response: As suggested by the reviewer, previous relevant studies are now reported in detail. (New second § of Introduction, as well as extension of next § where our previous lure study is described).

Attentional capture by lures: the hypotheses and the conclusions drawn from the results rely to a large extent on the question of which lures produce attentional capture. The authors suggest that a colour-matching lure should be more salient and produce more capture than a category-matching lure. It is pretty safe to assume that participants used the colour as the selection feature and not the identity. As such, it is reasonable to assume that the colour would produce goal-driven attentional capture. I think this can be stated clearly (and put in the context of previous studies about attentional capture and the effects of lures).
The hypothesis that attentional capture by lures that are irrelevant but share the target’s color would reduce accuracy is supported by previous studies (Folk et al., 2002, 2008; Moore & Weissman, 2010, 2011; Zivony & Lamy, 2014, 2016). However, while the authors assume that attentional capture would have a spatially specific effect, these studies found that disruptive effect is not spatial (or at least not entirely spatial), as it didn’t matter if the capturing non-target was in the same stream as the target (Folk et al., 2008, Zivony & Lamy, 2016) or in a different stream (Folk et al., 2002; Moore & Weissman, 2010; Zivony & Lamy, 2014). Instead, it was interpreted that the presence of a cue can trigger an AB and disrupt the triggering of attentional engagement to subsequent targets. The larger disruption by colour lure (indicated, for example the reduced accuracy on T1) coupled with an absence of location-specific effects can surely be explained in this context (though the location specific effect of digit lures could not). Moreover, given that the lures themselves were temporally distant from T2, it is possible that their presence reduced the T2-locked N2pc and accuracy by means of increasing the time it took to process T1 (thereby increasing the depth of the blink created by T1).

References:

Folk, C. L., Leber, A. B., & Egeth, H. E. (2002). Made you blink! Contingent attentional capture produces a spatial blink. Perception & psychophysics64(5), 741-753.
Folk, C. L., Leber, A. B., & Egeth, H. E. (2008). Top-down control settings and the attentional blink: Evidence for nonspatial contingent capture. Visual Cognition16(5), 616-642.
Moore, K. S., & Weissman, D. H. (2010). Involuntary transfer of a top-down attentional set into the focus of attention: Evidence from a contingent attentional capture paradigm. Attention, Perception, & Psychophysics72(6), 1495-1509.
Moore, K. S., & Weissman, D. H. (2011). Set-specific capture can be reduced by pre-emptively occupying a limited-capacity focus of attention. Visual Cognition19(4), 417-444.
Zivony, A., & Lamy, D. (2014). Attentional engagement is not sufficient to prevent spatial capture. Attention, Perception, & Psychophysics76(1), 19-31.
Zivony, A., & Lamy, D. (2016). Attentional capture and engagement during the attentional blink: A “camera” metaphor of attention. Journal of experimental psychology: human perception and performance42(11), 1886.

Response: Thank you for these insights and for providing this literature. The correct way would be to integrate this literature in the Introduction. But this appears difficult at this stage of the paper. Thus, we integrated this literature in the final conclusions:

After “Blue letters, more than black digits, might have led participants to believe that the blue digit had already been presented, thereby terminating their search for the true T2” we now continue:

“This reasoning may account for the minor role of spatial specificity of color-lure effects. In fact, similar conclusions have emerged from a research tradition where lures were not similar to T2 but rather to the only target that had to be detected in a central RSVP stream, and lure-type distractors were placed in the periphery of this central stream: Also here, color lures in the periphery had spatially non-specific negative effects, by interfering with detection of the centrally presented target [60-65].”

Results section readability: This is a complex design with numerous analyses, and together with the abbreviated condition names, makes the results section somewhat difficult to follow. Some of the phrasing throughout the results section can be improved to make the reading more accessible. I would prefer to see the details about each ANOVA right before the results themselves, as I found myself going back and forth from methods to results.

Response: Done as suggested, for identification rates of T1 and T2. (Such changes were not necessary with the ERP analyses because ANOVA designs had been described already in the first version).

Another possible suggestion, which may or may not be realistic, is to present a figure that is collapsed across conditions that are less relevant to the interpretation of the main findings (given that the full results are already reported in the table).

Response: In line with this suggestion to provide more structured results we compiled the ANOVA results in tables, following a suggestion made further below by this reviewer.

Results validity: As the authors mention themselves in the discussion, the study may have problems of power, given the numerous conditions and small amount of trials per condition. This can lower the readers’ confidence in the validity and replicability of the results. For example, while the multiple comparisons between each data point and 0 is indeed the easiest way to interpret the effects of lures, it also raises the concern of alpha inflation, especially for results whose p-value are close to .04. Moreover, given that many of the interaction effects were significant in ways that were not predicted by the authors (many of them only barely significant), it is very hard to conclude which of them reflect a real effect and which is spurious. This can be remedied in another behavioural experiment which aims to replicate the main results.

Response: In line with this suggestion, we reduced the number of tests (by eliminating the 32 t tests) and reduced the number of arcane interactions by simplifying the design (by focusing on the lag 3 data, thereby replacing the 3-level T1-T2 Relation factor by the 2-level Other-Target-Side factor).

Conducting another behavioral experiment is difficult because, first, the deadline granted to us for revising this paper by this journal’s editorial office is extremely tight (two weeks!) and, second, the first author has retired in June 2017. Therefore, his lab does not exist anymore.

N2pc results: when testing the N2pc for lures, the N2pc for the 2nd and 3rd lures may be reduced because of an AB effect (as reported by numerous previous studies, including from the main authors’ group as well as in other studies). This makes the analysis of the 2nd and 3rd lure less informative. Moreover, given that attention was already shifted towards the first lure, the negativity may reflect SSVEPs towards the attended stream rather than a real N2pc (e.g., Müller et al., 1998). This would explain why the 2nd lure has a different latency. To reject this explanation, the authors can show that the non-lure distractors do not produce an enhanced negativity.

Response: The reviewer’s point is convincing. Indeed, already close inspection of Figure 3 shows the fluctuations presumably caused by the sequence of distractors. We changed analysis, description, and discussion of results accordingly.

If the authors wish to show latency differences, I would prefer to see an analysis of latency onset rather than analysis of an early time window that was selected by looking at the results. In general, I think the N2pc analysis should be in the same time window throughout the manuscript, as different time windows raises the readers’ concern that they were tailored to the results. Regarding the lures, can the authors specify which of lures N2pc were different from 0?

Response: Lure-evoked N2pc and T2-evoked N2pc reach their peak at about 250 ms. Therefore, a unitary analysis window at 200-300 ms makes sense for both events, which is what we did now.
In contrast, T1-evoked N2pc reaches its peak at about 220 ms. An analysis window at 200-300 ms would not be appropriate. Therefore, we kept the 175-275 ms window as before.

Reference:
Müller, M. M., Picton, T. W., Valdes-Sosa, P., Riera, J., Teder-Sälejärvi, W. A., & Hillyard, S. A. (1998). Effects of spatial selective attention on the steady-state visual evoked potential in the 20–28 Hz range. Cognitive Brain Research, 6(4), 249-261.

Discussion:

Inhibition account: given that digit lures and colour lures produced the same enhanced negativity in FCz, and the finding that the effect of digit lures is generally positive, what evidence is there for the inhibition account? At the very least at seemed that enhanced negativity from one lure to another does not reflect enhanced negativity.

Response: Prompted by this comment, we shortened our evaluation of the frontal gating hypothesis in the present data from “This remains a possibility and would extend previous findings …” to the comment “This does not seem very probable”.

As the authors state, the complex pattern of results does not allow for a clear conclusion about the effects of lures. Here, as well, an additional behavioral experiment may be beneficial. For example, to test the possibility that the lures are used to predict the targets’ temporal position, two different blocks of temporal predictability vs. no predictability can be introduced.

Response: As stated above, conducting another behavioral experiment is difficult because, first, the deadline granted us for revising this paper by this journal’s editorial office is extremely tight (two weeks!) and, second, the first author has retired in June 2017 and his lab, therefore, does not exist anymore.

Minor issues:

Introduction:

the authors could go into more details about why the study is important.

Response: Done, due to responses required by this reviewer’s above questions and by Reviewer 2’s first question.

I’m not sure that the lures used in this study reflect time-dependent changes of relevance.

Response: This passage is about lures in general in RSVP tasks. Such lures have been identical to T2 targets except for the fact that they were presented too early. Therefore, the change from such lures to T2 reflects time-dependent changes of relevance. The reviewer is right that this does not strictly apply to lures used in the present study because of the special feature of the present study that T2 is defined by the combination of two characteristic features whereas lures possess one of those features only. But this does not invalidate the statement about lures in general.

The N2pc does not reflect attentional shifting per se, but rather a downstream attentional process (Kiss, Van Velzen & Eimer, 2008; Zivony, Allon, Luria & Lamy, 2018), and can occur even when no attentional shifting is required. It would be helpful to present the evidence that the N2pc is smaller when attention was already captured to the target’s side. This is also an issue that affects the conclusions that can be drawn from the N2pc findings.

Response: Many thanks for shifting our attention to the Zivony et al. (2018) paper! After studying it carefully, I must say that I am not fully convinced. Please, also see our similar cue-papers (Åšmigasiewicz, Westphal, Verleger et al., once in JoCN 2015, once in Neuropsychologia 2017). We might discuss this in more depth outside of this review procedure.

Method:
1. If I understand correctly, the targets were much brighter than the contralateral distractors. This should be addressed as a limitation, as the effects may be related to perceptual imbalances rather than attentional imbalances.

Response: This paper is about the effects of lures on targets, and the targets remained constant across lure conditions. Therefore, we do not see why a constant target feature is a limitation. The particular problems in titrating task difficulty created by our use of a color-digit T2 (rather than our usual black-digit T2) have been already addressed in the previous version (4.1.1).

Personally I found the boldface font somewhat distracting. Perhaps Italics would be better?

Response: According to this suggestion, we changed boldface to non-italics while the other characters denoting the sequence are now consistently written in italics. (We were not entirely consistent with the use of italics before. The very first sequences had not been written in italics while most of the following ones were. Thus, prompted by this comment, we standardized this to have italics as the usual font for denoting sequences and highlighting targets in the sequence by the absence of italics.)

Results:

In table 1: the headline should be T2, rather than target.
Response: “target” in the headline has been changed to “T1 (top half) and T2 (bottom half)”. It will be helpful to have a decimal added to the averages reported …

Response: Being an obsessive reader, I experience much interference by unnecessary decimals, so even though I followed the reviewer’s suggestion I am not happy with it.

and SEs (preferably within-subject SEs) instead of SDs, which can ease the interpretation of the table.
Response: When acting as a reviewer myself, I use to require SDs rather than SEs, because what is relevant is the empirical variation in the sample rather than the hypothetical variation of the sample mean when the experiment would be replicated. So this seems to be a clash of philosophies.
Relieving the impact of this clash was that we followed a suggestion made by Reviewer 3 who wrote that this table seems to be rather unnecessary and who, therefore, suggested to make it a supplementary table in the appendix. Which we did.

In figure 2, lower panels: perhaps add a dotted line on 0 to help reader orient themselves.

Response: Done.

Sometimes lag 1 and lag 3 are written as lag1 and lag3.

Response: Changed as suggested.

Given that many of the effects were significant, the authors may consider using a table to detail the results of the main ANOVA for lure effects on T1 and T2, and then use the main text to give further details and refer to the table.

Response: We gladly followed this suggestion.

“In this ANOVA, the constant term was different from zero … indicating that N2pc was overall reduced by the presence of lures” – I don’t understand how the ANOVA’s constant leads to that conclusion…

Response: Description of this result has been improved: “indicating that N2pc was overall reduced by the presence of lures” has been changed to “indicating that there was an overall N2pc difference between lure and no-lure trials which, as Figure 6 shows, was an overall reduction of N2pc by the presence of lures.”

Reviewer 2 Report

This manuscript presents a systematic investigation into the effects of lures that appear prior to target stimuli in the attentional blink paradigm. The lures were either semantically similar (lures and T2 were both numbers) or visually similar (lures and T2 were both blue). The authors also reported EEG ERP components corresponding to lures and targets. I appreciate the authors’ efforts to clearly explain a relatively high number of conditions. I think that this work is of specific interest to those interested in the attentional blink. However, I have concerns about the experimental design and the robustness of the effects given the limited number of subjects and noisy patterns of behavior.

In the semantic priming condition, one of the three lures was identical to the target (T2). The authors justify this choice by saying that, in the other condition, the lure color matched the target color. However, unlike the color match, these identical lures could aid the discrimination task. That is, in the semantic priming task, the subjects had two opportunities to encode the target digit.

Further clarify this design choice and the possible ramifications. More prominently note this feature of the experimental design when introducing semantic priming. In the results, address whether this could account for differences between color and digit conditions.

I am concerned about the robustness of the behavioral findings, especially given a relatively small number of participants and trials per condition. In particular, the T1 identification results appear noisy even when there were no lures and the findings related to specific sides (T2 right side vs. T1 left side). For example, subtracting high (though not significantly different) performance seems to drive most of the decrement in performance in the R3R condition in Figure 2 bottom left.

State explicitly whether the obtained sample size (14) was determined a priori did not seem to be determined a priori, and report post-hoc power obtained for this sample size State explicitly how many trials remained in each condition on average after artifact rejection I recommend collecting additional data from an equal number of participants, at least behaviorally, to replicate the observed behavioral findings.

Other concerns:

I accept the choice to exclude targets on the same side at lag 1 from the design. However, I was concerned about the imbalance across the experiment (indeed, the presence of lures could indicate to participants to attend to the other side) and whether lag 1 effects should be included in many analyses, given that this generates an imbalance: many more trials include a side switch. Could the authors clarify when lag 1 analyses are included in the statistics, and whether results change if these conditions are excluded from analyses?

The EOG artifact rejection only described eye movements towards the first lure. Given that the ERPs are lateralized, the authors should remove any brief eye movements towards the target/lure side. If that was the case, please clarify in methods. If that was not the case, redo more comprehensive artifact rejection or provide justification why this is unnecessary.

N2PC results. The N2PC for the second and third color lures has a peculiar form, which could have impacted the latency findings. How were the epochs for early vs. middle determined? If the middle epoch was centered around the peak of the first color lure, would the earlier onset result for the second and third lures still hold?

Minor comments:

In Figure 6, as I understand it, the left figure is the same side as T1, and the filled lines refer to the lures appearing on the same side as the T2. I would recommend clarifying that.

The introduction could be updated to foreshadow the frontal ERP results. As written, the results imply that were prior expectations for this ERP.

Some simplification of language in the results section may facilitate digestion of the results. I think the first sentence of the discussion is not grammatically correct.

Author Response

Reviewer 2

This manuscript presents a systematic investigation into the effects of lures that appear prior to target stimuli in the attentional blink paradigm. The lures were either semantically similar (lures and T2 were both numbers) or visually similar (lures and T2 were both blue). The authors also reported EEG ERP components corresponding to lures and targets. I appreciate the authors’ efforts to clearly explain a relatively high number of conditions. I think that this work is of specific interest to those interested in the attentional blink. However, I have concerns about the experimental design and the robustness of the effects given the limited number of subjects and noisy patterns of behavior.

Our general response: We thank this reviewer very much for his/her insightful comments. In fact, I have rarely encountered such constructive and insightful comments from all contributing reviewers as in the present case.

In the semantic priming condition, one of the three lures was identical to the target (T2). The authors justify this choice by saying that, in the other condition, the lure color matched the target color. However, unlike the color match, these identical lures could aid the discrimination task. That is, in the semantic priming task, the subjects had two opportunities to encode the target digit.

Further clarify this design choice and the possible ramifications. More prominently note this feature of the experimental design when introducing semantic priming. In the results, address whether this could account for differences between color and digit conditions.

Response: Prompted by this question, we described (among other studies) Harris et al.’s and our previous study in more detail, and then referred to those results when stating the hypotheses:

“In Harris et al.’s study [12] a stream of object drawings was presented at screen center. Participants had to identify the two red objects (T1 and T2) among the other, black objects. The black object that was presented two frames before T1 could be the same object as the red T2. These lures had negative effects on T2 identification when T2 was presented briefly after T1. …”

“Specifically, in our task [15] lures had negative effects on T2 identification when the lag between T1 and T2 was 3 frames, This effect was shifted towards positive priming when lures and T2 were in the same stream and when the intervening T1 was ‘out of the way’ in the other stream and when one of three lures was identical to T2. This latter result apparently differs from the above-reported results by Niedeggen et al. and Harris et al. [4-9,12] where identity of lures and T2 impeded rather than facilitated T2 identification. …”

“To optimize the presumed distinction of effects, one of the three digit lures in a trial was identical to T2, like in half the lure trials of our previous study [15]. Positive priming from lures on T2 might be considered trivial when one of the lures is T2. But negative identity priming from lures on T2 has been shown as well [12], and, moreover negative priming from identical irrelevant to relevant stimuli when separated by masks or other stimuli has often been demonstrated [16-18].

I am concerned about the robustness of the behavioral findings, especially given a relatively small number of participants and trials per condition. In particular, the T1 identification results appear noisy even when there were no lures and the findings related to specific sides (T2 right side vs. T1 left side). For example, subtracting high (though not significantly different) performance seems to drive most of the decrement in performance in the R3R condition in Figure 2 bottom left.

Response: Prompted by this comment, we inserted an entirely new section in the Discussion where we review previous results of ours. (Sorry that this increased our number of self-citations). Briefly, a T1 right-side advantage has been found in studies of ours on 163 participants, and a null result in studies on 123 participants. A same-side advantage (i.e., T1-followed-by-same-side-T2) has been found in studies on 73 participants, a null result with 206 participants, and one study on 50 participants even yielded the opposite effect.

We conclude in that section that “the present good identification of right T1 followed by right T2 seems to be a both replicable and variable phenomenon.”

We then continue: “The advantage of right T1 might be related to left-hemisphere specialization for language [48], and its variability across experiments may confirm Hellige’s [49] conclusion that laterality of letter identification depends on the detailed circumstances. Accounting for the effect of side of the following T2 on identification of the previous T1 seems more challenging. Worse T1 identification when T2 is on the same side [47] might be due to backward masking. The opposite better T1 identification when T2 is on the same side [14,15,40,46] may be taken to suggest that, like a post-cue [50,51], T2 is able to draw attention not only to its ongoing stimulus stream but, given beneficial circumstances, also to the short-term memory representation of previous stimuli in this stream.”

State explicitly whether the obtained sample size (14) was determined a priori did not seem to be determined a priori, and report post-hoc power obtained for this sample size

Response: The sample size was determined a priori, to match the sizes of our previous studies. We added in Section 2.1: “This sample size was well in the range of our previous studies using this task where consistent behavioral and ERP effects had been obtained.”
With special regard to T1 RVF advantage: With this advantage having been obtained in studies on totally 163 participants and no effect having been obtained in studies on totally 123 participants: Unambiguously deciding on this issue seems to require a sample size of hundreds of participants, which is not realistic.

State explicitly how many trials remained in each condition on average after artifact rejection

Response: We added the following statements: In Section 3.1 (behavior): “Each data point was computed from 25 trials per participant.” In Section 3.2 (lure-evoked ERPs): “Lure-evoked ERPs were computed from 150 trials per participant (minus artifact-affected epochs). In Section 3.3 (lure effects on T1 processing): “T1-evoked ERPs were computed from about 120 trials per participant (150 trials per cell minus about 20% trials with incorrect responses) minus artifact-affected trials.“ In Section 3.4 (lure effects on T2 processing: “T2-evoked ERPs were computed from 28 trials per participant (50 trials per cell minus about 33% trials with incorrect responses minus artifact-affected trials).”

I recommend collecting additional data from an equal number of participants, at least behaviorally, to replicate the observed behavioral findings.

Response: Conducting another behavioral experiment is difficult because, first, the deadline granted to us for revising this paper by this journal’s editorial office is extremely tight (two weeks!) and, second, the first author has retired in June 2017 and his lab, therefore, does not exist anymore.

Other concerns:

I accept the choice to exclude targets on the same side at lag 1 from the design. However, I was concerned about the imbalance across the experiment (indeed, the presence of lures could indicate to participants to attend to the other side) and whether lag 1 effects should be included in many analyses, given that this generates an imbalance: many more trials include a side switch. Could the authors clarify when lag 1 analyses are included in the statistics, and whether results change if these conditions are excluded from analyses?

Response: Prompted by this suggestion and following a suggestion explicitly made by Reviewer 3, we excluded lag 1 from most analysis, in order to simplify the presentation of results. With identification rates, we then added an ancillary analysis that compared lag 1 data with lag 3 data.

The EOG artifact rejection only described eye movements towards the first lure. Given that the ERPs are lateralized, the authors should remove any brief eye movements towards the target/lure side. If that was the case, please clarify in methods. If that was not the case, redo more comprehensive artifact rejection or provide justification why this is unnecessary.

Response: Here are the grand-average waveforms of hEOG contralateral minus ipsilateral to lures.

(see attachment - I hope the upload will work)

Therefore, we inserted in Methods: “Including those two participants, the grand average waveforms of hEOG deviations toward lures reached a maximum of about 1 µV at 400 ms after onsets of 1st and 2nd lures, and much smaller values after 3rd lures, with 1 µV corresponding to average eye movements of about 0.07° towards the lure stream, which still appeared to be an acceptable level.”

N2PC results. The N2PC for the second and third color lures has a peculiar form, which could have impacted the latency findings. How were the epochs for early vs. middle determined? If the middle epoch was centered around the peak of the first color lure, would the earlier onset result for the second and third lures still hold?

Response: We followed the reviewer’s suggestion and centered analysis around the peak of N2pc evoked by the first color lure. Prompted by this reviewer’s critical remarks, we were more ready to accept Reviewer 1’s proposal that the fluctuations seen with 2nd and 3rd lures were not necessarily N2pcs (with earlier onset) but rather increases of N1 components evoked by distractors following the attention-catching lures. We changed our description of results and analysis of these results accordingly.

Minor comments:

In Figure 6, as I understand it, the left figure is the same side as T1, and the filled lines refer to the lures appearing on the same side as the T2. I would recommend clarifying that.

Response: We clarified this by changing the figure captions accordingly. Likewise in Figure 2.

The introduction could be updated to foreshadow the frontal ERP results. As written, the results imply that were prior expectations for this ERP.

Response: It is not entirely clear what this criticism refers to. Because we had clearly stated in the Introduction: “it may be assumed that such frontal negativity will not build up with the semantic digit lures in the present study but will do so with the distracting salient color lures”. Maybe this had been phrased too cautiously. Therefore, in response to this criticism, we changed this to read “we assume for the present study that such frontal negativity will not build up with the same-category digit lures but will do so with the distracting salient color lures.” [“Semantic” was changed to “same alphanumeric category” or simply “same-category” in response to a point made by Reviewer 3].

Some simplification of language in the results section may facilitate digestion of the results.

Response: The entire passage about T1 identification and T2 identification was rewritten.

I think the first sentence of the discussion is not grammatically correct.

Response: The first two sentences of the Discussion have been rephrased.

Reviewer 3 Report

The authors investigate the effect of lures with the same color or alphanumerical categorical on perceptual identification performance. The lures preceded two targets that were separated by 1 or three frames of 110 ms. The first target, T1, was a letter shown in red. The second target, T2, was a digit shown in blue. T1 and T2 were shown in separate RSVP streams on the left and right. T2 identification performance showed that lures with the same alphanumerical category facilitated performance, at least when shown in the RSVP stream on the opposite side. However, the electrophysiological results were not consistent with the behavioral results. The N2pc to the lures showed a larger N2pc for color, but only for the first lure. Potentials at frontal sites showed no marked difference between color and digit lures. The N2pc to T1 showed a marked difference between same and opposite side color lures that is not really matched by better identification of T1 in this situation. There were no differences between same and opposite-side digit lures. Finally, there were no differences in the N2pc to T2, which is puzzling because of the large differences in identification performance.

Although the results are inconclusive, the study presents an interesting research question. The main problem I see is that the results are difficult to understand because of the way they are presented.

It is not clear why a large Table with all the cells is presented at the beginning of the results section. The table is not motivated any further and could be deleted or transferred to the appendix.

The relevant ERP analysis focus on the SOA with three frames. Given that the authors have reported results from this study many times, it may be ok to drop the SOA with one frame from the analysis of behavioral effects. This would increase the readability, because a straightforward ANOVA model could be used for the behavioral data: T1 position (left, right) x T2 position (left, right), which would cover L3L, L3R, R3L, R3R. Results of this analysis may be easier to understand than a main effect of T1-T2 relation, which reflects an imbalanced design. If the authors do not want to drop effects at the first SOA, they could add a second ANOVA with T1/T2 position (left/right, right/left) x time (SOA 1, SOA 2), which would cover L1R, L3R, R1L, L3R. This ANOVA would give information about effects of time and position for contralateral T1/T2.

Here are some more suggestions for improving the readability.

Line 523: “But actually, color lures’ negative effects on T2 identification were only slightly, non-significantly smaller when T2 was in the same stream than when T2 was in the other stream.” This conclusion would be much better supported by the alternative ANOVA model that I suggested.

Line 22 : The relation between letters and digits is not semantic. Semantic relations concern category membership or the like (i.e., fruit, vehicle, etc.). I think it would be more appropriate to talk about alphanumerical category. Similarity of lures refers to being drawn from the same alphanumeric category as T2. Semantic similarity requires word stimuli.

Line 280 : The results are difficult to follow. I suggest that the authors specify the ANOVA-model at the beginning of each section by enumerating factor names and factor levels. For instance “A one-way ANOVA (T1-T2 relation: L1R, R1L, L3L, R3R, L3R, R3L)”. Otherwise, it is hard to know what is meant by “T1-T2 relation”.

Line 284: The sentence starting with “was resolved in pair-wise comparisons to differences between same-side …” is very hard to understand and may not be grammatical (“comparisons to differences”?). Please reword. Maybe it would be good to mention the condition names (L1R, etc.).

Line 357: What is meant by “position numbers” ? Lure number? Serial position? I cannot be spatial position.

Lines 357 – 363: It is not clear to me how the presented effects are evidence for a larger N2pc with color lures. In particular, what is a “general effect across position numbers in the late epoch”? How does this relate to the larger N2pc with color? Overall, I found this section difficult to understand.

Line 356: From looking at the Figure, it seems wrong to conclude that the N2pc was larger with color lures. In addition, there was no main effect of lure type. Only the N2pc to the first color lure was bigger. For the remaining serial positions, there was no difference, just some variation in the timing. It would be interesting to run an ANOVA just on the digit lures to see whether there are any effects at all. If so, the interactions can all be attributed to the first color lure and the earlier onset of the second color lure.

Line 386: “Lure Number” would be better “Serial Position of Lure” because “number” is a very equivocal term.

Line 457: What is meant by “constant term”? The conclusion is that the presence of lures reduces the N2pc, but the effect of lure (presence, absence) was not mentioned on lines 456-457.

Line 536: “semantic priming” seems to be the wrong word here, as the difference was one in alphanumerical category.

Line 576: “Pd” should be “PD” with a subscript D.

Lines 574-587: The main reason why no PD occurred in the present study should be sought in the similarity between lures and targets. It is true that studies on the PD used mostly simultaneous presentation. However, when target and distractor were drawn from the same dimension and therefore related, there was an N2pc to the distractor (Hilimire, Mounts, Parks, & Corballis, 2011; Liesefeld, Liesefeld, Töllner, & Müller, 2017). A PD is typically observed with targets that are easy to find and clearly distinct from the distractor (Burra & Kerzel, 2014; Gaspar & McDonald, 2014; Hickey, Di Lollo, & McDonald, 2009). When targets are difficult to find (Barras & Kerzel, 2017) or the target is ambiguous because of nontarget-target swaps (Burra & Kerzel, 2014), an N2pc will result.

Signed Dirk Kerzel

Barras, C., & Kerzel, D. (2017). Salient-but-irrelevant stimuli cause attentional capture in difficult, but attentional suppression in easy visual search. Psychophysiology, 54(12), 1826-1838. doi:10.1111/psyp.12962

Burra, N., & Kerzel, D. (2014). The distractor positivity (Pd) signals lowering of attentional priority: Evidence from event-related potentials and individual differences. Psychophysiology, 51(7), 685-696. doi:10.1111/psyp.12215

Gaspar, J. M., & McDonald, J. J. (2014). Suppression of salient objects prevents distraction in visual search. Journal of Neuroscience, 34(16), 5658-5666. doi:10.1523/JNEUROSCI.4161-13.2014

Hickey, C., Di Lollo, V., & McDonald, J. J. (2009). Electrophysiological indices of target and distractor processing in visual search. Journal of Cognitive Neuroscience, 21(4), 760-775. doi:10.1162/jocn.2009.21039

Hilimire, M. R., Mounts, J. R., Parks, N. A., & Corballis, P. M. (2011). Dynamics of target and distractor processing in visual search: evidence from event-related brain potentials. Neuroscience Letters, 495(3), 196-200. doi:10.1016/j.neulet.2011.03.064

Liesefeld, H. R., Liesefeld, A. M., Töllner, T., & Müller, H. J. (2017). Attentional capture in visual search: Capture and post-capture dynamics revealed by EEG. Neuroimage, 156(Supplement C), 166-173. doi:10.1016/j.neuroimage.2017.05.016

Author Response

Reviewer 3

The authors investigate the effect of lures with the same color or alphanumerical categorical on perceptual identification performance. The lures preceded two targets that were separated by 1 or three frames of 110 ms. The first target, T1, was a letter shown in red. The second target, T2, was a digit shown in blue. T1 and T2 were shown in separate RSVP streams on the left and right. T2 identification performance showed that lures with the same alphanumerical category facilitated performance, at least when shown in the RSVP stream on the opposite side. However, the electrophysiological results were not consistent with the behavioral results. The N2pc to the lures showed a larger N2pc for color, but only for the first lure. Potentials at frontal sites showed no marked difference between color and digit lures. The N2pc to T1 showed a marked difference between same and opposite side color lures that is not really matched by better identification of T1 in this situation. There were no differences between same and opposite-side digit lures. Finally, there were no differences in the N2pc to T2, which is puzzling because of the large differences in identification performance.

Although the results are inconclusive, the study presents an interesting research question. The main problem I see is that the results are difficult to understand because of the way they are presented.

Our general response: We thank this reviewer very much for his insightful comments. In fact, I have rarely encountered such constructive and insightful comments from all contributing reviewers as in the present case.

It is not clear why a large Table with all the cells is presented at the beginning of the results section. The table is not motivated any further and could be deleted or transferred to the appendix.

Response: As suggested, we transferred the table to the appendix.

The relevant ERP analysis focus on the SOA with three frames. Given that the authors have reported results from this study many times, it may be ok to drop the SOA with one frame from the analysis of behavioral effects. This would increase the readability, because a straightforward ANOVA model could be used for the behavioral data: T1 position (left, right) x T2 position (left, right), which would cover L3L, L3R, R3L, R3R. Results of this analysis may be easier to understand than a main effect of T1-T2 relation, which reflects an imbalanced design. If the authors do not want to drop effects at the first SOA, they could add a second ANOVA with T1/T2 position (left/right, right/left) x time (SOA 1, SOA 2), which would cover L1R, L3R, R1L, L3R. This ANOVA would give information about effects of time and position for contralateral T1/T2.

Response: We followed this suggestion, reanalyzed T1 identification and T2 identification and rewrote the entire results section accordingly.

Here are some more suggestions for improving the readability.

Line 523: “But actually, color lures’ negative effects on T2 identification were only slightly, non-significantly smaller when T2 was in the same stream than when T2 was in the other stream.” This conclusion would be much better supported by the alternative ANOVA model that I suggested.

Response: Right. Indeed, now this effect is at the p = .05 threshold.

Line 22 : The relation between letters and digits is not semantic. Semantic relations concern category membership or the like (i.e., fruit, vehicle, etc.). I think it would be more appropriate to talk about alphanumerical category. Similarity of lures refers to being drawn from the same alphanumeric category as T2. Semantic similarity requires word stimuli.

Response: Good point. “Semantic” has been changed throughout to “same-category” or “alphanumeric category” or “category”.

Line 280 : The results are difficult to follow. I suggest that the authors specify the ANOVA-model at the beginning of each section by enumerating factor names and factor levels. For instance “A one-way ANOVA (T1-T2 relation: L1R, R1L, L3L, R3R, L3R, R3L)”. Otherwise, it is hard to know what is meant by “T1-T2 relation”.

Response: Done as suggested, for identification rates of T1 and T2. (Such changes were not necessary with the ERP analyses because ANOVA designs had been described already in the first version).

Line 284: The sentence starting with “was resolved in pair-wise comparisons to differences between same-side …” is very hard to understand and may not be grammatical (“comparisons to differences”?). Please reword. Maybe it would be good to mention the condition names (L1R, etc.).

Response: All these suggestions make sense. But due to the reviewer’s suggestion for changing the ANOVA design, this passage has been replaced in its entirety.

Line 357: What is meant by “position numbers” ? Lure number? Serial position? I cannot be spatial position.

Response: We changed this throughout to “serial position”, thanks to the reviewer’s comment at l.386.

Lines 357 – 363: It is not clear to me how the presented effects are evidence for a larger N2pc with color lures. In particular, what is a “general effect across position numbers in the late epoch”? How does this relate to the larger N2pc with color? Overall, I found this section difficult to understand.

Response: Following suggestion by all reviewers, analysis of lure-evoked N2pc has been completely revised.

Line 356: From looking at the Figure, it seems wrong to conclude that the N2pc was larger with color lures. In addition, there was no main effect of lure type. Only the N2pc to the first color lure was bigger. For the remaining serial positions, there was no difference, just some variation in the timing. It would be interesting to run an ANOVA just on the digit lures to see whether there are any effects at all. If so, the interactions can all be attributed to the first color lure and the earlier onset of the second color lure.

Response: We ran the analyses suggested by the reviewer and included the results in the manuscript. The reviewer’s intuitions were not confirmed in this case.

Line 386: “Lure Number” would be better “Serial Position of Lure” because “number” is a very equivocal term.

Response: As mentioned above, we changed this throughout to “serial position”, thanks to this comment.

Line 457: What is meant by “constant term”? The conclusion is that the presence of lures reduces the N2pc, but the effect of lure (presence, absence) was not mentioned on lines 456-457.

Response: The sentence was extended to better convey its meaning: “In this ANOVA on difference values, the constant term was different from zero, F1,13 = 7.2, p = .02, indicating that there was an overall N2pc difference between lure and no-lure trials which, as Figure 6 shows, was an overall reduction of N2pc by the presence of lures.”

Line 536: “semantic priming” seems to be the wrong word here, as the difference was one in alphanumerical category.

Response: Cf. above, the reviewer’s comment at line 22. Accordingly, this was also changed here to read “priming of digits as a category”.

Line 576: “Pd” should be “PD” with a subscript D.

Response: Changed as suggested by the reviewer.

Lines 574-587: The main reason why no PD occurred in the present study should be sought in the similarity between lures and targets. It is true that studies on the PD used mostly simultaneous presentation. However, when target and distractor were drawn from the same dimension and therefore related, there was an N2pc to the distractor (Hilimire, Mounts, Parks, & Corballis, 2011; Liesefeld, Liesefeld, Töllner, & Müller, 2017). A PD is typically observed with targets that are easy to find and clearly distinct from the distractor (Burra & Kerzel, 2014; Gaspar & McDonald, 2014; Hickey, Di Lollo, & McDonald, 2009). When targets are difficult to find (Barras & Kerzel, 2017) or the target is ambiguous because of nontarget-target swaps (Burra & Kerzel, 2014), an N2pc will result.

Response: This makes much sense. We replaced our tentative explanation by this one.

Signed Dirk Kerzel

Barras, C., & Kerzel, D. (2017). Salient-but-irrelevant stimuli cause attentional capture in difficult, but attentional suppression in easy visual search. Psychophysiology, 54(12), 1826-1838. doi:10.1111/psyp.12962

Burra, N., & Kerzel, D. (2014). The distractor positivity (Pd) signals lowering of attentional priority: Evidence from event-related potentials and individual differences. Psychophysiology, 51(7), 685-696. doi:10.1111/psyp.12215

Gaspar, J. M., & McDonald, J. J. (2014). Suppression of salient objects prevents distraction in visual search. Journal of Neuroscience, 34(16), 5658-5666. doi:10.1523/JNEUROSCI.4161-13.2014

Hickey, C., Di Lollo, V., & McDonald, J. J. (2009). Electrophysiological indices of target and distractor processing in visual search. Journal of Cognitive Neuroscience, 21(4), 760-775. doi:10.1162/jocn.2009.21039

Hilimire, M. R., Mounts, J. R., Parks, N. A., & Corballis, P. M. (2011). Dynamics of target and distractor processing in visual search: evidence from event-related brain potentials. Neuroscience Letters, 495(3), 196-200. doi:10.1016/j.neulet.2011.03.064

Liesefeld, H. R., Liesefeld, A. M., Töllner, T., & Müller, H. J. (2017). Attentional capture in visual search: Capture and post-capture dynamics revealed by EEG. Neuroimage, 156(Supplement C), 166-173. doi:10.1016/j.neuroimage.2017.05.016